# Bacteriophage targeting microbiota alleviates non-alcoholic fatty liver disease induced by high alcohol-producing *Klebsiella pneumoniae*

Lin Gan[1], Yanling Feng[1], Bing Du[1], Hanyu Fu[1], Ziyan Tian[1], Guanhua Xue[1], Chao Yan[1], Xiaohu Cui[1], Rui Zhang[1], Jinghua Cui[1], Hanqing zhao[1], Junxia Feng[1], Ziying Xu[1], Zheng Fan[1], Tongtong Fu[1], Shuheng Du[1], Shiyu Liu[1], Qun Zhang[1], Zihui Yu[1], Ying Sun[2] ✉ & Jing Yuan[1] ✉

Our previous studies have shown that high alcohol-producing *Klebsiella pneumoniae* (HiAlc *Kpn*) in the intestinal microbiome could be one of the causes of non-alcoholic fatty liver disease (NAFLD). Considering antimicrobial resistance of *K. pneumoniae* and dysbacteriosis caused by antibiotics, phage therapy might have potential in treatment of HiAlc *Kpn*-induced NAFLD, because of the specificity targeting the bacteria. Here, we clarified the effectiveness of phage therapy in male mice with HiAlc *Kpn*-induced steatohepatitis. Comprehensive investigations including transcriptomes and metabolomes revealed that treatment with HiAlc *Kpn*-specific phage was able to alleviate steatohepatitis caused by HiAlc *Kpn*, including hepatic dysfunction and expression of cytokines and lipogenic genes. In contrast, such treatment did not cause significantly pathological changes, either in functions of liver and kidney, or in components of gut microbiota. In addition to reducing alcohol attack, phage therapy also regulated inflammation, and lipid and carbohydrate metabolism. Our data suggest that phage therapy targeting gut microbiota is an alternative to antibiotics, with potential efficacy and safety, at least in HiAlc *Kpn*-caused NAFLD.

Non-alcoholic fatty liver disease (NAFLD) is one of the commonest chronic liver diseases globally[1], which includes simple fatty liver disease and non-alcoholic steatohepatitis (NASH). NAFLD can result in liver fibrosis, cirrhosis and hepatocellular carcinoma if without proper treatments[2]. According to the statistics, there is about 25% of the global population suffering from NAFLD, while the incidence of NASH could increase 56% in the next 10 years[3,4]. It has been reported that numerous factors might be associated with NAFLD, including obesity, insulin resistance syndrome and the gut microbiota[5,6]. Our

previous data derived a Chinese cohort have revealed that 60% of NAFLD patients carry high alcohol-producing *Klebsiella pneumoniae* (HiAlc *Kpn*), which could be one of the causes of NAFLD[7]. Overgrowth of HiAlc *Kpn* might be the underlying cause of NAFLD because the bacteria can ferment carbohydrates into alcohol in the gut. The alcohol produced by HiAlc *Kpn* (≥30 mmol/L) within the gut microbiota can be delivered to the liver through the portal vein system, leading to hepatocyte steatosis and lipid metabolic disorder, which result in NAFLD[8,9]. Such NAFLD caused by HiAlc *Kpn*

[1]Department of Bacteriology, Capital Institute of Pediatrics, 100020 Beijing, China. [2]Department of Immunology, School of Basic Medical Sciences, Capital Medical University, 100069 Beijing, China. ✉e-mail: ying.sun@ccmu.edu.cn; yuanjing6216@163.com

has been named as "endogenous alcohol fatty liver disease (endo-AFLD)"[7].

Unfortunately, there is no approved pharmacotherapy available for NAFLD. Current studies have mainly focused on targeting hepatic fat accumulation, oxidative stress, inflammation, apoptosis, the intestinal microbiome, metabolic endotoxemia and hepatic fibrosis. The potential drugs include de novo lipogenesis inhibitors, apoptosis signaling kinase 1 inhibitors, macrolide antibiotics and galectin 3 antagonists[10,11]. Our previous data have also shown that usages of bacteriophage (phage) or certain antibiotics to selectively eradicate HiAlc *Kpn* can prevent steatohepatitis in the experimental murine models[7]. These suggest that targeting HiAlc *Kpn* might be an effective therapy for endo-AFLD. However, the antibiotic resistance of *K. pneumonia* is a serious problem in clinic practice, while the most HiAlc *Kpn* strains are multidrug-resistant (MDR) bacteria. According to a multi-center clinical study in China, *K. pneumoniae* accounts for 66.7% of carbapenem-resistant *Enterobacteriaceae* collected from 25 hospitals in 14 provinces[12].

The efficacy of the known therapies based on combination with antibiotics aminoglycosides, colistin, fosfomycin and tigecycline is unsatisfactory, while new antibiotics targeting MDR *K. pneumonia* are still in development[13,14]. In addition, it has been shown that gut microbiota plays important role in liver diseases and lipid metabolism through the gut–liver axis. Therefore, imbalance in the intestinal flora induced by antibiotics might lead to unknown and unexpected side effects[15,16].

It has been shown that bacteriophage therapy has been proposed as a solution to combat infections caused by MDR bacteria[17]. In principle, phages are ubiquitous and highly strain-specific, and thus can potentially be used to edit microbiota. Furthermore, the positive effect of lytic phage has been verified in the clinic[18–20]. For example, usage of phages targeting *K. pneumoniae* associated with inflammatory bowel disease is able to effectively inhibit growth of the bacteria, attenuate inflammation and relieve disease severity[21]. A recent study has further shown that treatment with bacteriophages specifically targeting cytolytic *Enterococcus faecalis* decreases cytolysin in the liver and abolishes ethanol-induced liver disease in humanized mice[20]. Assessment of *K. pneumoniae*-targeted phages in the artificial human gut and healthy volunteers has also demonstrated resilience, safety and viability of such treatment[21]. To further verify the potential efficacy of removing etiological HiAlc *Kpn* for alleviating of endo-AFLD, the established murine model of steatohepatitis was employed in the present study to clarify the effectiveness, side effects and molecular mechanism of phage for treatment of endo-AFLD in vivo. Data showed that HiAlc *Kpn*-specific phage attenuated hepatic dysfunction through reprogramming the gut microbiota, which possibly aids to reveal the molecular mechanism of phage treatment for endo-AFLD, and provides a unique perspective for clinic.

## Results

### Genomic and biological characteristics of a lytic phage against HiAlc *Kpn*

HiAlc *Kpn* was defined as the causative agent of NASH accompanied with recurrent pancreatitis (RP) of a male (50–60 years old, with no alcoholism) at Beijing Chaoyang Hospital, Capital Medical University. Interestingly, metagenomic analysis showed that the abundance of *K. pneumoniae* in his intestinal microbiota was approximately inversely proportional to the abundance of *Klebsiella* phages (Fig. S1a, b). Furthermore, the quantity of *K. pneumoniae* in the patient's microbiome was significantly increased at the inflammatory phase of the patient with NASH in the absence of antibiotic treatment, while the quantity of *Klebsiella* phage remained relatively low. In contrast, the opposite trend was observed in recovery periods of the patient.

To explore whether the presence of HiAlc *Kpn* and phages was associated with the clinical characteristic, 16 subjects from our

previous cohort, who were with NASH but without alcoholism, were further investigated and followed-up. HiAlc and MedAlc (medium alcohol-producing, which produces 20-30 mmol/L alcohol) *Kpn* strains were isolated from fecal samples of these patients before weight loss using metagenomic analysis (Supplementary Data 1). The data revealed that the quantity of *K. pneumoniae* strains was significantly reduced in these NASH followed-up patients compared to the same patients before weight loss, while the quantity of *Klebsiella* phages was increased in gut microbiota of a total of 6 patients (Fig. S1c). Taken together, these results suggest that there should be an equilibrium in the abundances between *K. pneumoniae* and *Klebsiella* phages in vivo. These data also raised a possibility that the phage might alleviate inflammation via its self-regulation, which might be a potential natural medication for treatment of endo-AFLD.

To investigate whether *Klebsiella* phage specifically targets HiAlc *Kpn* but not affects other microbiota in vivo, we screened *Klebsiella* phages by using 86 strains of HiAlc *Kpn* isolated from clinical NAFLD patients as host bacteria. Results showed that a lytic phage phiW14 (CGMCC No. 23085, GenBank No. OK655936, which specifically targets HiAlc *Kpn* HK1) had the strongest cleavage ability and the most extensive host spectrum. Thus, the phage phiW14 and HiAlc *Kpn* HK1 were chosen for further experiments.

HiAlc *Kpn* HK1 (ST1536, which produces 49.7 mmol/L alcohol) was initially isolated from a NASH/RP patient, having similar biological characteristics to HiAlc *Kpn* TH1 (ST1536, which produces 60.8 mmol/L alcohol, GCA_001676825.1), presenting multidrug resistant but imipenem susceptible, and causing NAFLD through high-producing alcohol[7]. Furthermore, the phage phiW14 formed clear and large plaques on a lawn of strain HK1 (Fig. 1a). Transmission electron microscopy (TEM) indicated that phiW14 possesses a 50-nm diameter icosahedral head and a 20-nm non-contractile tail and belongs to the *Podoviridae* family viruses (Fig. 1a). The one-step growth curve showed that the latent period of phiW14 was 10 min, the burst time was 80 min, and the average burst size was 129 plaque-forming units (PFUs)/ cell (Fig. 1b). The optimal multiplicity of infection (MOI) was 0.001 of phiW14 vs HK1. At MOI 10 to 0.0001, phiW14 completely inhibited HiAlc *Kpn* HK1 (Fig. 1c). Phage phiW14 was able to infect 18 of 86 HiAlc *Kpn* strains at MOI 0.001, which belong to ST25, ST65, ST86, ST193, ST375, ST1536, and three new STs (Supplementary Data 2). Also, phiW14 was most stable at pH 5 to 10 (Fig. 1d) and at 4 °C to 40 °C (Fig. 1e), which could be tolerant of the environmental stresses. Whole genome sequence showed that phiW14 is a circular double stranded DNA of 50,247 bp with 79 putative open reading frames, but without genes of resistance-drug or virulence (Fig. 1f and Supplementary Data 3). In addition, a total of 52 proteins of phiW14 were identified (Supplementary Data 4), and 26 proteins were classified into different biological functions by Cluster of Orthologous Groups of Proteins (COG) analysis, including biosynthesis, transcription, mobilome, transport and metabolism-related proteins (Fig. 1g). Phylogenetic clustering of large terminase proteins clearly distinguished phiW14 from the other phage families (Fig. 1h). Taken together, these results suggest that phiW14 might be a potential candidate for treatment of endo-AFLD.

### Specially eradicating HiAlc *Kpn* in fecal microbiota prevented steatohepatitis in recipient mice

To clarify the effect of phage treatment on endo-AFLD, mice were gavaged with the HiAlc *Kpn* HK1 pretreated with either phage phiW14 or phikp15 (HK1/phiW14-fed mice or HK1/phikp15-fed mice) (Fig. 2a), in which phage phikp15 was used as negative control because it cannot lyse HK1 in vitro. In addition, using HiAlc *Kpn* strains HK2 (ST447, which produces 42.5 mmol/L alcohol) and HK3 (ST101, which produces 41.5 mmol/L alcohol) (both was isolated from a patient with NASH) as bacterial host, phages phikp16 (targeting HK2) and phikp17 (targeting HK3) were obtained from the patient's feces. As for FMT mice, fecal

samples from the patient mentioned above were also proceed with the same pretreatment subsequently conducted to be transplanted in FMT/Phage mice (Fig. 2b).

Compared with mice fed with chow diet, HK1 or EtOH-fed, FMT mice treated with fecal microbiota carrying HiAlc *Kpn* HK2 and HK3

had steatohepatitis and obviously pathological changes in the livers. The contents of *K. pneumoniae* in feces (Fig. 2c, d), and as serological indicators, including levels of alanine transaminase (ALT), aspartate transaminase (AST), triglyceride (TG), diamine oxidase (DAO) and D-lactate (D-LA) levels (Fig. 2f), were significantly

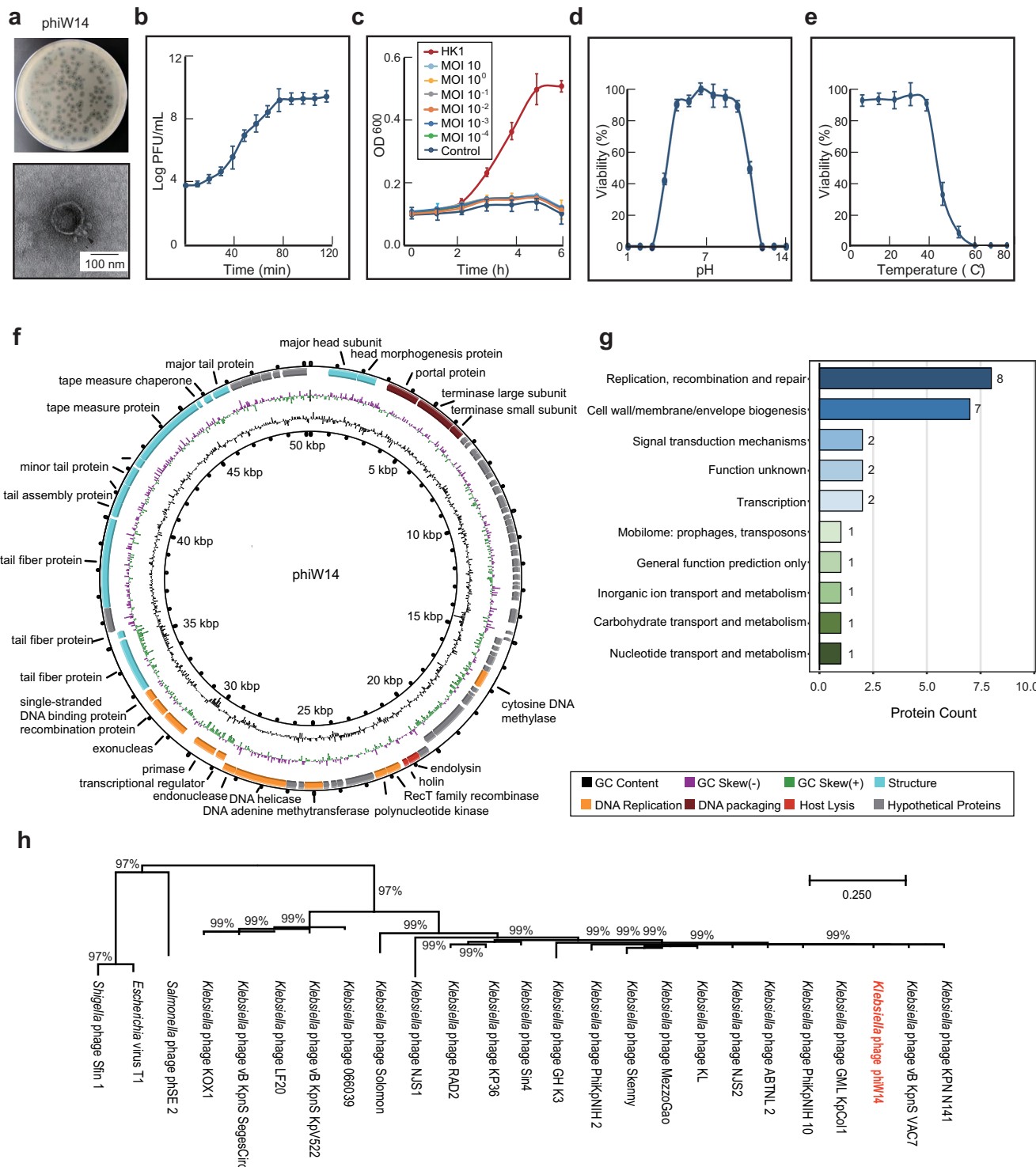

**Fig. 1 | Genomic and biological characteristics of phage phiW14. a** Morphology and plaques of phage phiW14. **b** One-step growth curve of phage phiW14 ($n = 4$). **c** Kill curve of phage phiW14 against high alcohol-producing *Klebsiella pneumoniae* (HiAlc *Kpn*) HK1 ($n = 5$). **d** Stability of phage phiW14 under pH conditions ($n = 4$). **e** Stability of phage phiW14 under different temperatures ($n = 4$). **f** Genomic map of

phage phiW14. **g** Cluster of Orthologous Groups of Proteins classification of phage phiW14. **h** Phylogenetic tree based on terminase large subunit amino acid sequences of phage phiW14. Values are expressed as the mean ± SD ($n = 6$). Source data are provided as a Source data file.

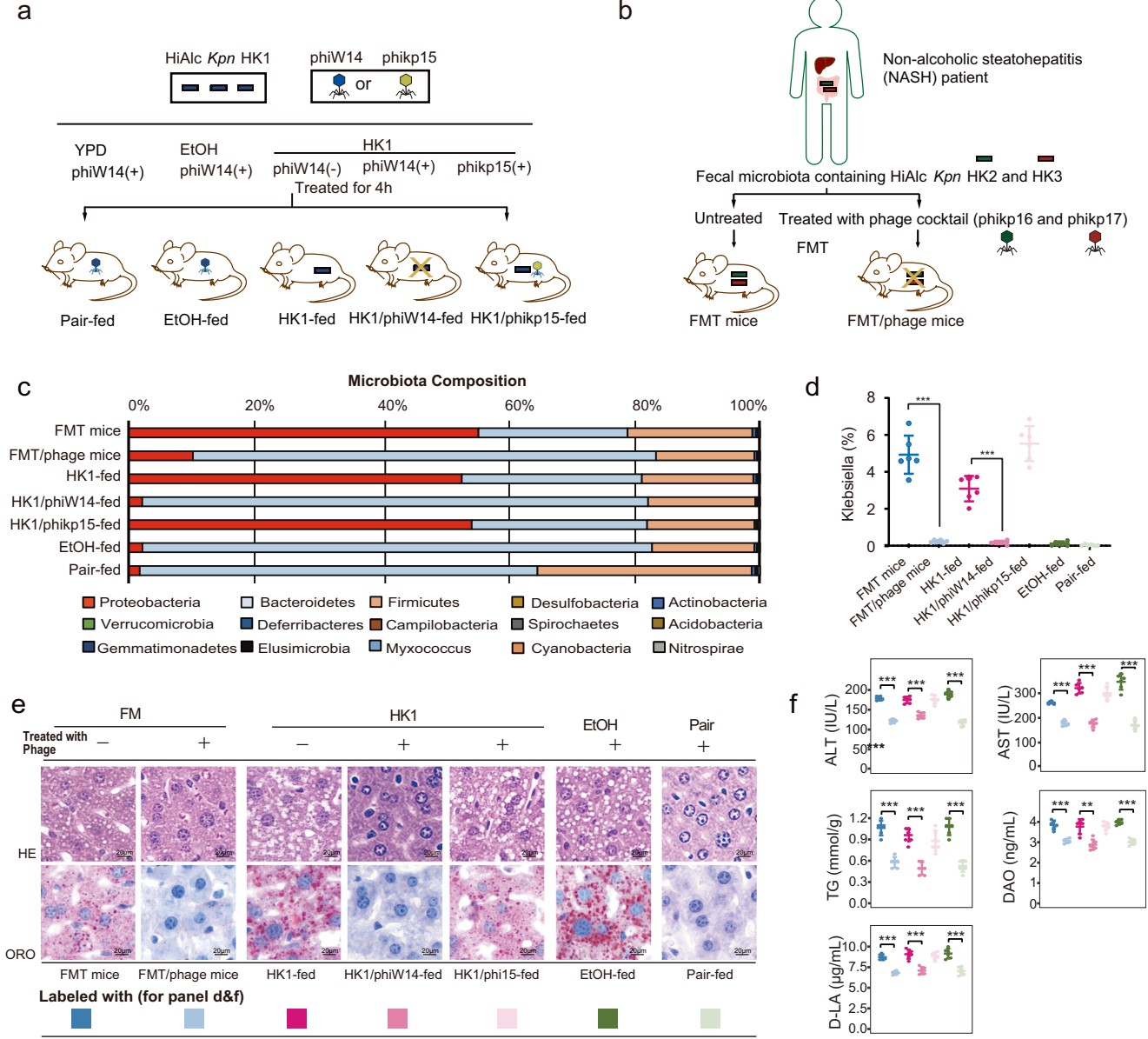

**Fig. 2 | HiAlc *Kpn*-fed mice and fecal microbiota transplantation (FMT) following phage pretreatment in vitro. a** Constructions of experimental mice fed with Pair, EtOH, HK1, HK1/phiW14 and HK1/phikp15. **b** Constructions of FMT and FMT/phage murine models. **c** Compositions of phylum-based intestinal microbiota in feces of experimental mice. **d** Abundance of *Klebsiella* in feces of experimental mice (*n* = 6, *P* < 0.0001). **e** Hematoxylin−eosin (HE) and Oil Red O (ORO) stainings of sections of liver tissues of experimental mice. **f** Concentrations of alanine transaminase (ALT), aspartate transaminase (AST), triglyceride (TG), diamine oxidase (DAO), and D-lactate (D-LA) in serum of experimental mice (*n* = 6, except the *P* value of DAO levels between the group HK1-fed and HK1/ phiW14-fed is 0.001, other *P* values < 0.0001). Values are expressed as the mean ± SD (*n* = 6 mice/group). One-way ANOVA, *P* value < 0.01 (**) or 0.001 (***). Source data are provided as a Source data file.

elevated in mice fed with HiAlc *Kpn*, HK1/phikp15, FMT, and EtOH, accompanying hepatic steatosis (Fig. 2e). However, there were no abnormalities observed in the mice either fed with HK1/phiW14 or fed with FMT/phage, compared with pair-fed mice. In addition, the overall composition of the intestinal microbiome at the phylum level was not affected by the phage treatment (Fig. 2c). These suggest that treatment with phage could prevent steatohepatitis possibly through specially eradicating HiAlc *Kpn* in fecal microbiota of recipient mice.

## Bacteriophage therapy reprogrammed the gut microbiota without obvious side effects

To explore whether there are some unexpected side effects, mice with steatohepatitis induced by HiAlc *Kpn* HK1 were treated with phage phiW14 at the range of $10^4$, $10^5$, and $10^6$ PFUs for 1, 4 and 7 days. The results showed that abundance of *Klebsiella* was effectively inhibited by all three concentrations of phage at all time points in vivo (Fig. 3a, b). However, in the pathological point of view, significantly alleviated hepatic steatosis was only observed in the mice treated with phage continuously for one week (Fig. 3c). In addition, serological indicators, including levels of ALT, AST, TG, DAO and D-LA, were also significantly decreased in mice of the groups with phage therapy for one week (Fig. 3d). It was noted that there was no significant effect in the mice with phage treatment for1 or 4 days in vivo.

To further evaluate the side effects of phage therapy in vivo, hepatic and renal function-related indicators and histopathological changes of liver, kidney, and intestine tissues were also examined. The

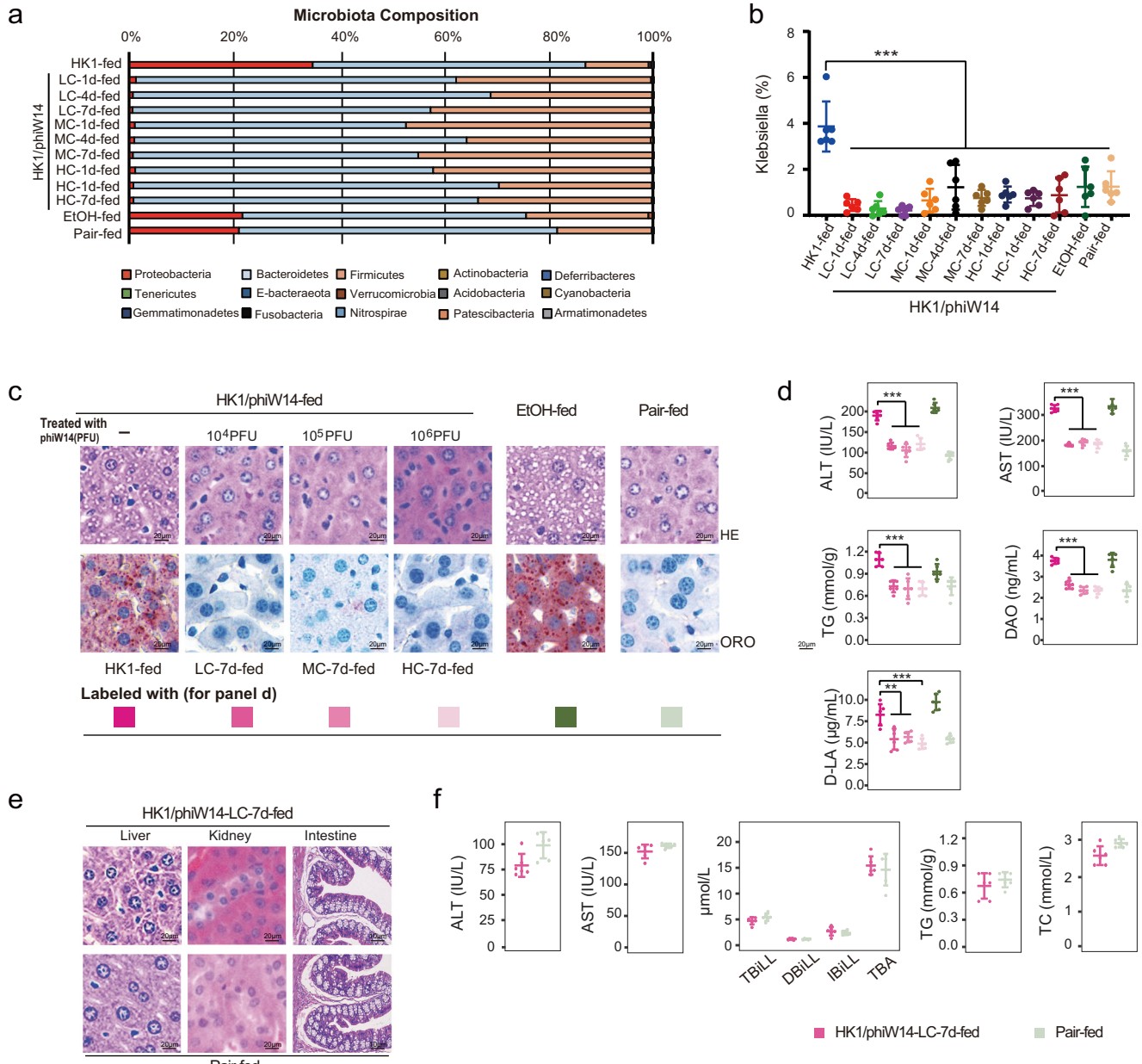

**Fig. 3 | Bacteriophage reprograms gut microbiota of mice with HiAlc *Kpn*-induced steatohepatitis, but without side effects.** Mice with steatohepatitis (gavaged with HiAlc *Kpn* for 4 weeks) were treated with phage phiW14 at 10⁴ (low concentration, LC), 10⁵ (median concentration, MC), and 10⁶ (high concentration, HC) PFUs for 1, 4 and 7 days. **a** Compositions of phylum-based intestinal microbiota in feces of experimental mice. **b** *Klebsiella* concentrations in feces of experimental mice (*n* = 6, *P* < 0.0001). **c** HE and ORO stainings of sections of liver tissues of experimental mice. **d** Concentrations of ALT, AST, TG, DAO, and D-LA in serum of experimental mice (*n* = 6, except the *P* value of D-LA levels between the group HK1-

fed and LC-7d-fed is 0.003, and the *P* value of D-LA levels between the group HK1-fed and MC-7d-fed is 0.001, other *P* values < 0.0001). **e** HE staining of sections of liver, kidney and intestine tissues of mice treated with 10⁴ PFUs of phiW14. **f** Concentrations of ALT, AST, total bilirubin (TBil), direct bilirubin (DBil), indirect bilirubin (IBil), total bile acid (TBA), TG and total cholesterol (TC) in serum of mice treated with 10⁴ PFUs of phiW14 (*n* = 6). Values are expressed as the mean ± SD (*n* = 6 mice/group). One-way ANOVA, *P* value < 0.01 (**) or 0.001 (***). Source data are provided as a Source data file.

data indicated that there was no obvious pathological injury such as cell degeneration in the phage-treated mice (Fig. 3e). Compared with pair-fed mice, there were no significant changes either in levels of ALT, AST, total cholesterol (TC), TG, total bilirubin (TBil), direct bilirubin (DBil), indirect bilirubin (IBil) or in level of total bile acid (TBA) of the experimental mice treated with phage. These suggest that treatment with phage phiW14 could reverse fatty liver by scavenging HiAlc *Kpn* (Fig. 3f). Thus, all mice with HiAlc *Kpn*-induced steatohepatitis were treated with 10⁴ PFU of phage for one week in subsequent experiments.

## Bacteriophage targeting of HiAlc *Kpn* alleviated steatohepatitis in vivo

To determine the mechanism by which phage alleviates endo-ALFD, mice with steatohepatitis were constructed through gavaging HiAlc *Kpn* for 4 or 8 weeks. The experimental mice were then fed with phage phiW14 or imipenem for further one week (Fig. 4a). As expected, hepatic steatosis, injury, inflammation and the blood alcohol concentrations (BAC) were relieved in the 4- and 8-week models of mice treated either by the phage or the antibiotic (Fig. 4b–d). Phage and antibiotic therapies also reduced indexes of injured liver function,

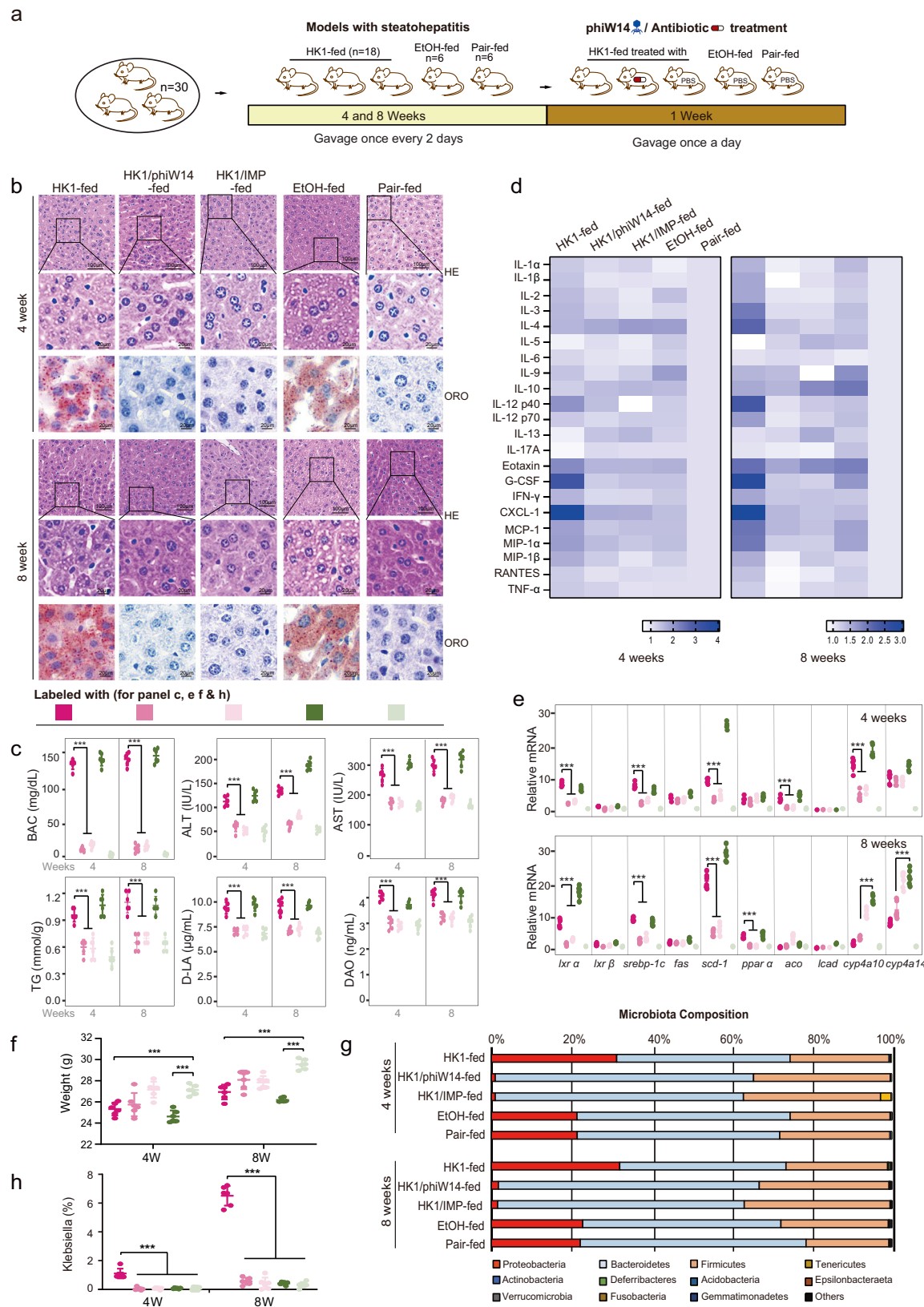

including levels of ALT, AST, DAO, D-LA and TG, although some indicators such as ALT showed no significantly difference between the mice with steatohepatitis and the mice treated with phage- or antibiotic (Fig. 4c). The expression of 22 cytokines such as IL-1β, IL-6, IL-10 and TNF-α (Fig. 4d), 5 lipogenic genes including *Srebp-1c* and *Scd-1*, and 5 lipolytic genes (such as *Pparα* and *Cyp4a14*) (Fig. 4e) were also

regulated after treatment with phage or antibiotic. These data suggest that phage therapy might be effective for the treatment of HiAlc *Kpn*-caused NAFLD. In addition, both HiAlc *Kpn* and alcohol feeding resulted in weight loss in mouse models (*P* < 0.001). Interestingly, although there was no statistically significant difference in body weight between groups of HK1- and HK1/phiW14-fed mice, the average weight

**Fig. 4 | Bacteriophage targeting of HiAlc *Kpn* alleviates steatohepatitis in vivo.** **a** Constructions of Pair-, EtOH-, HK1-, HK1/phiW14- and HK1/IMP-fed murine models. Mice were gavaged with strain HK1, EtOH or YPD broth for 4 weeks or 8 weeks, and then treated with phage phiW14, antibiotic imipenem or PBS for a further week. After the construction of murine models, mice were euthanized for subsequent analysis. **b** HE and ORO stainings of sections of liver tissues of experimental mice. **c** Concentrations of blood alcohol, ALT, AST, TG, DAO, and D-LA in serum of experimental mice ($n = 6$, $P < 0.0001$). **d** Concentrations of cytokines in liver tissues of experimental mice. **e** Relative expression levels of lipogenic genes in liver tissues of experimental mice ($n = 6$, $P < 0.0001$). **f** Body weight of experimental mice ($n = 6$, $P < 0.0001$). **g** Compositions of phylum-based intestinal microbiota in feces of experimental mice. **h** Abundance of *Klebsiella* in feces of experimental mice ($n = 6$, $P < 0.0001$). Values are expressed as the mean ± SD ($n = 6$ mice/group). One-way ANOVA, $P$ value < 0.001 (***). Source data are provided as a Source data file.

of group of mice fed with HK1/phiW14 (25.83 g in 4 weeks, 28.14 g in 8 weeks) was also higher than that of group of mice fed with HK1 (25.40 g in 4 weeks, 27.02 g in 8 weeks). These suggest that phage treatment might improve mice health to a certain extent.

Furthermore, although the abundance of *Klebsiella* was decreased significantly in the mice treated by both the phage and the antibiotic, the microbial community structures were different between the two treatments (Fig. 4g, h). Obvious changes in the overall composition of the fecal microbiome were observed in mice treated with imipenem. In 4-week model of mice with HiAlc *Kpn*-induced steatohepatitis, treatment with imipenem reduced the abundance of *Bacteroidetes* and *Proteobacteria*, but increased the abundance of *Tenericutes*. In 8-week models, the absolute abundance of *Firmicutes* and *Proteobacteria* was decreased in the mice treated with imipenem (though the relative abundance was increased). Thus, long term treatment with imipenem caused changes in intestinal flora, and even resulted in dysbacteriosis although such treatment also cleared HiAlc *Kpn* and alleviated steatohepatitis.

### Transcriptional and metabolic reprogramming induced by phage therapy in vivo

We further analyzed transcriptome and metabolome profiles in the livers of all groups. The results showed that mice with HiAlc *Kpn*-induced steatohepatitis had significantly changes after phage or antibiotic therapy. After phage treatment, muscle synthesis-associated pathways of mice with steatohepatitis were upregulated, while pathways of inflammation or apoptosis, such as the Wnt signaling pathway and extracellular signal-regulated protein kinase 1 cascades were downregulated (Fig 5a, b and Supplementary Data 5). In contrast, the treatment with imipenem upregulated biological process-associated pathways, but downregulated inflammation and apoptosis pathways, including leukocyte migration and mitogen-activated protein kinase (MAPK) cascades (Fig 5c, d and Supplementary Data 5).

In addition, amounts of the metabolites citraconic acid, α-tocopherol, and sebacic acid, which were enriched into glycerolipid metabolism pathways, as well as glycerol 3-phosphate of the glycerophospholipid metabolism pathways were increased, while L-lactic acid in the glycolysis and gluconeogenesis pathways were decreased in mice treated with HK1/phiW14-fed, as compared with mice with HiAlc *Kpn*-induced steatohepatitis (Fig. 6a–e and Supplementary Data 6). Furthermore, metabolites including 2,2,2-trichloroethanol, digalacturonic acid, and UDP-glucuronic acid, which were enriched in pentose and glucuronate interconversions, ascorbate and aldarate metabolism, and metabolism of xenobiotics by cytochrome p450 pathways were upregulated in the of mice fed with HK1/IMP (Fig. 6f–j and Supplementary Data 6). These data suggest that phage therapy has efficacy on mice with HiAlc *Kpn*-induced steatohepatitis, possibly through not only targeting HiAlc *Kpn*, but also regulating inflammation, lipid metabolism and carbohydrates metabolism.

### Bacteriophage therapy efficiently relieved HiAlc *Kpn*-induced steatohepatitis in germfree mice

The effect of phage therapy was further verified in germfree mice (Fig. 7a) gavaged by intestinal flora as mentioned in Fig. 2b. Similar with the effect of phage in SPF mice, the intestinal flora from the patient with NASH caused steatohepatitis in germfree mice, and *K. pneumoniae* colonization in gut was observed by scanning electron

microscope (SEM). After phage treatment, the related indicators of steatohepatitis such as 16S *rRNA* sequencing (Fig. 7b, c), pathological change (Fig. 7d) and serological indicators (Fig. 7e) were alleviated, while the *K. pneumoniae* strains were eliminated in the gut of germ-free mice.

### The efficacy of bacteriophage therapy in murine model of NASH

To explore the efficacy of phage in general NAFLD model, murine model of NASH induced by MCD (methionine-choline deficient) diet was also included in the present study. As shown in Fig. 8a, mice were randomized into 5 groups, including MCD-fed, MCD/HK1-fed (MCD-fed and HK1 gavaged), MCD/HK1/phiW14-fed (MCD-fed, HK1 and phiW14 gavaged), MCD/phiW14-fed (MCD-fed and phiW14 gavaged) and MCS (fed with methionine-choline-supplemented, as negative control). From the results of 16S *rRNA* sequencing (Fig. 8b, c), pathological changes (Fig. 8d) and serological indicators (Fig. 8e), phage treatment alleviated steatohepatitis in mice with MCD diet plus HiAlc *Kpn* feeding, which might reflect the important role of HiAlc *Kpn* in the pathogenic process of NASH.

In addition, we observed that the MCD/HK1 group had more severe pathological changes in hepatic steatosis and inflammation compared to the MCD group, and there was a noticeable occurrence of cellular phagocytosis in the MCD/HK1 group of 8-week model (Fig. 8d). The NAFLD activity scores (NAS) also showed that the MCD/HK1 group had a significantly higher score (Fig. 8e) than that of the MCD group in 8-weeks model ($P < 0.05$). However, there was no statistically significant difference in serological indicators between two groups. This is possibly due to that though the hepatic steatosis in both two groups has pathologically occurred, little change in the serum indicators can be observed until the disease progressed to a more severe stage. This, of course, needs to be verified in further research.

Phage treatment did not alleviated steatohepatitis in mice fed with MCD diet alone, this is probably pointing that there are diversities in the pathogenic mechanisms of NAFLD. On the other hand, these data also suggest that phage therapy might be highly specific, at least in our animal models of NASH.

## Discussion

It has been known that the intestine is able to affect the liver through connecting the portal venous system, biliary system, and mediators in the circulatory system. Therefore, because of receiving portal vein blood from the intestine, the liver is the first organ affected by dysbiosis of the gut microbiota and their metabolites. In addition to maintaining liver homeostasis, gut microbiota might also contribute to liver disease[22]. In this point of view, editing the gut microbiota may be an alternative for effective treatment of endo-NAFLD caused by HiAlc *Kpn*. Current antibiotic therapies, however, are not specific for *Klebsiella* and could cause side effects because they also remove beneficial commensal bacteria[23]. In addition, the antibiotic resistance of *K. pneumoniae* has been a serious global problem, while the most HiAlc *Kpn* strains are MDR bacteria.

Phage therapy has been applied to most common types of infection, including bacteremia, otitis, respiratory tract infections, urinary tract infections, skin and soft tissue infections, and gastrointestinal tract infections[19]. Bacteriophage therapy, as alternative stratagem to antibiotics, can effectively and specifically edit the intestinal microbiota[24,25]. Our data showed that there was a dynamic equilibrium

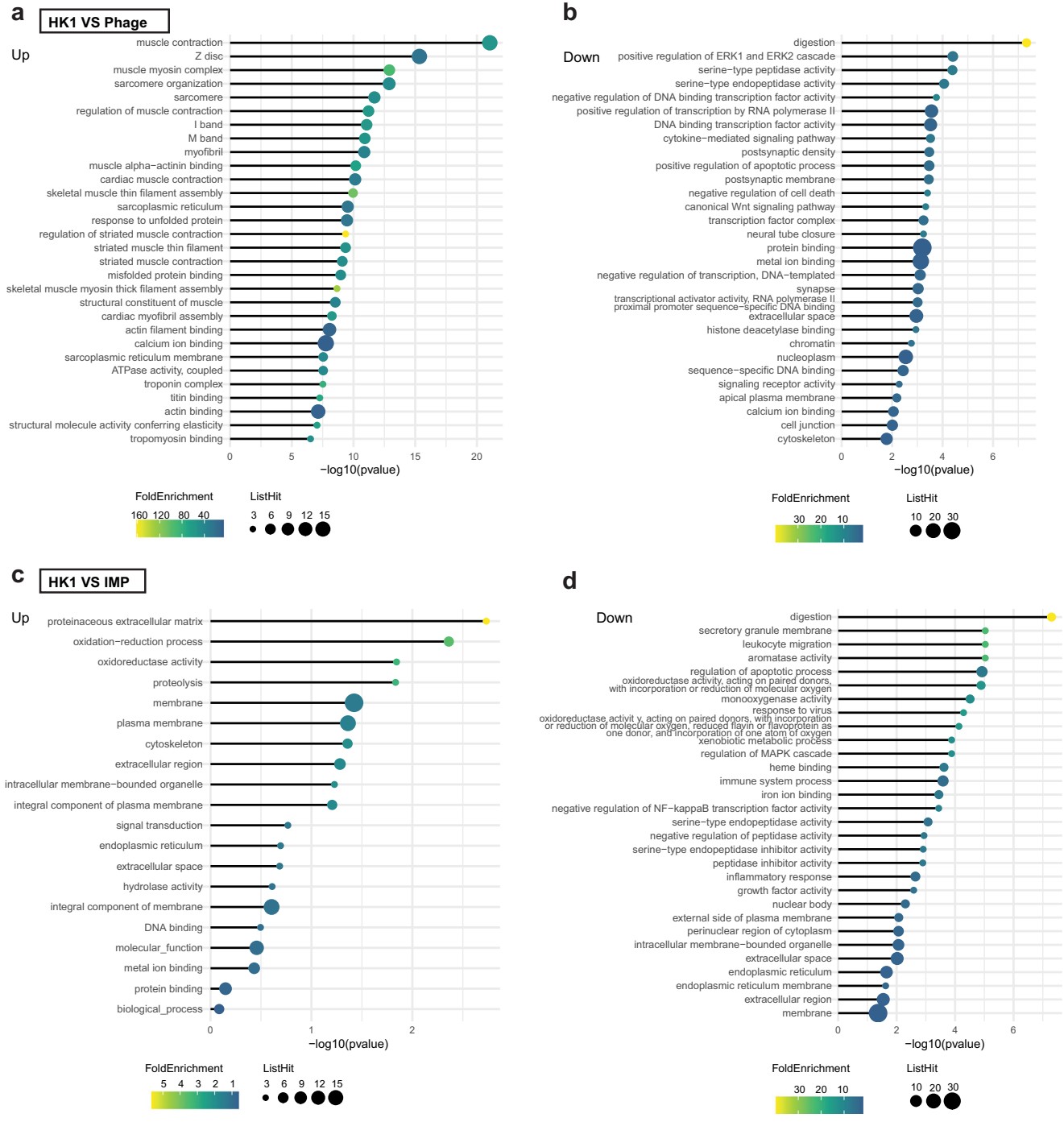

**Fig. 5 | Transcriptomic profiling of mice with HiAlc *Kpn*-induced steatohepatitis. a** Kyoto Encyclopedia of Genes and Genomes (KEGG) pathways enriched in upregulated DEGs; **b** KEGG pathways enriched in downregulated DEGs comparing groups of mice fed with HK1 and HK1/phiW14. **c** KEGG pathways enriched in upregulated DEGs; **d** KEGG pathways enriched in downregulated DEGs comparing groups of mice fed with HK1 and HK1/IMP.

in the abundance between *K. pneumoniae* and *Klebsiella* phages presented in feces of a patient with NASH, suggesting that phage may be a potential natural medication for treatment of endo-AFLD.

To clarify this concept, we screened a lytic phage phiW14 and explored the effects of the phage on HiAlc *Kpn*-caused endo-AFLD. The data showed that phiW14, without genes associated to lysogeny, antibiotic resistance or virulence, had a good ability to lyse HiAlc *Kpn* strain HK1, suggesting that it may be an effective phage for medical use with no obvious side effects. Supportingly, results further showed that treatment with the phage specifically targeting HiAlc *Kpn* attenuated hepatic steatosis, dysfunction, and immune disorders, but without

obvious side effects or overall changes in the composition of the gut microbiota in a murine model of steatohepatitis induced by HiAlc *Kpn*.

It is particularly interesting that expression of interleukin (IL)-10 was up-regulated in the liver tissues of mice treated with phage phiW14. It has been well known that IL-10 has an anti-inflammatory effect, and plays a protective role against hepatic apoptosis, necrosis and fibrosis[26]. Another study has also shown that the phages specifically targeting *Staphylococcus aureus* and *Pseudomonas aeruginosa* regulate IL-10 production via human mononuclear cells[27]. Collectively, our data suggest that the phage therapy could alleviates steatohepatitis, possibly through targeting HiAlc *Kpn* and enhancing IL-10

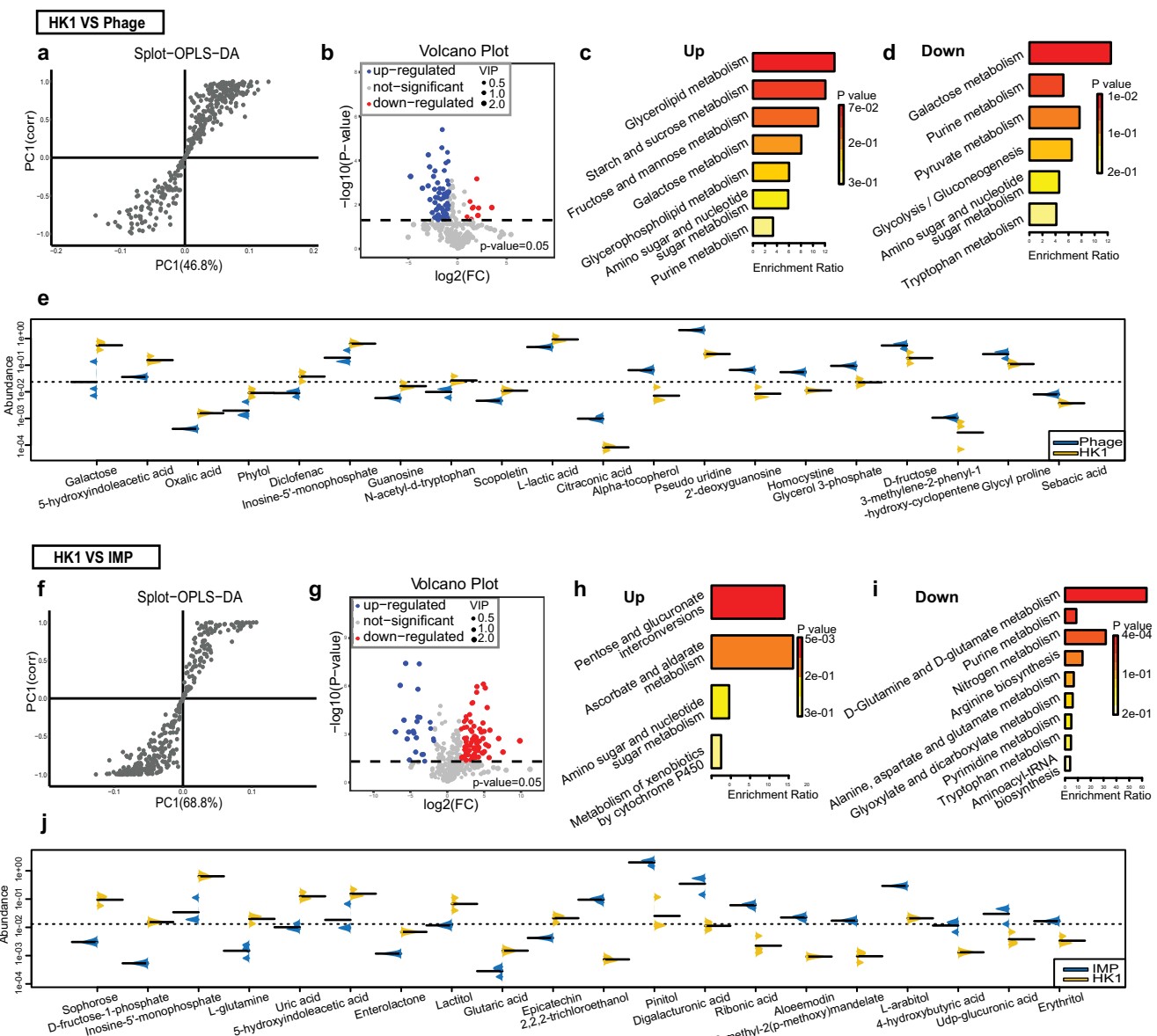

**Fig. 6 | Metabolomics profiles of mice with HiAlc *Kpn*-induced steatohepatitis.** **a** Orthogonal partial least-squares-discriminant analysis (OPLS-DA); **b** Volcano plot of differential metabolites; **c** KEGG pathways enriched in upregulated metabolites; **d** KEGG pathways enriched in downregulated metabolites; **e** differential metabolites between groups of mice fed with HK1 and HK1/phiW14. **f** OPLS-DA; **g** volcano plot of differential metabolites; **h** KEGG pathways enriched in upregulated metabolites; **i** KEGG pathways enriched in downregulated metabolites; **j** differential metabolites between groups of mice fed with HK1 and HK1/IMP. Values are expressed as the mean ± SD ($n = 6$ mice/group). Differential metabolites with $P$ value < 0.05 and VIP value > 1 were chosen for Metabo Analyst-based KEGG pathway enrichment analysis.

production. Notably, obviously relieved hepatic injury was only observed in the experimental mice treated by phage for 7 continuous days, but not in those treated for 1 or 4 days. Previous studies have shown that phage can be seen as a potential invader by the immune system and rapidly eradicated from systemic circulation[28]. These suggest that the relatively long-term usage of phage might be a key point for treatment of endo-AFLD. On the other hand, applications of phage cocktails, encapsulation and high-concentration phage may help to overcome this problem[29]. Further study of the pharmacokinetics of phage will be required.

Endogenous ethanol produced by HiAlc *Kpn* increases intestinal permeability, which leads to translocation of pathogen-associated molecular pattern molecules (PAMPs) to the liver[7,9,30]. PAMPs such as lipopolysaccharide (LPS) can translocate from the intestine via the portal vein into the liver. LPS can be recognized by pattern recognition

receptors expressed by relevant cells of the liver, which causes hepatic damage, liver fibrosis and inflammation. Similar pathways in addition to alcohol attack have been verified to contribute to the development of NAFLD[30,31]. In particular, activations of NOD-like receptor protein 3 (NLRP3) and NLRP6 have been identified as stimuli of hepatic inflammation, which cause caspase 1 activation, inflammatory cytokine production, and activations of nuclear factor-κB (NF-κB) and MAPK pathways, ultimately resulting in cell death and liver fibrosis[32,33]. The transcriptomic profiles of liver tissues of the experimental mice showed that both phage and antibiotic therapy decreased apoptotic- and inflammation-associated pathways, including regulations of NF-κB, MAPK and the extracellular-signal regulated kinase cascade, which are required for activating hepatic stellate cells)[34]. These suggest that the changes of above pathways might be countable to relaxation of endo-AFLD.

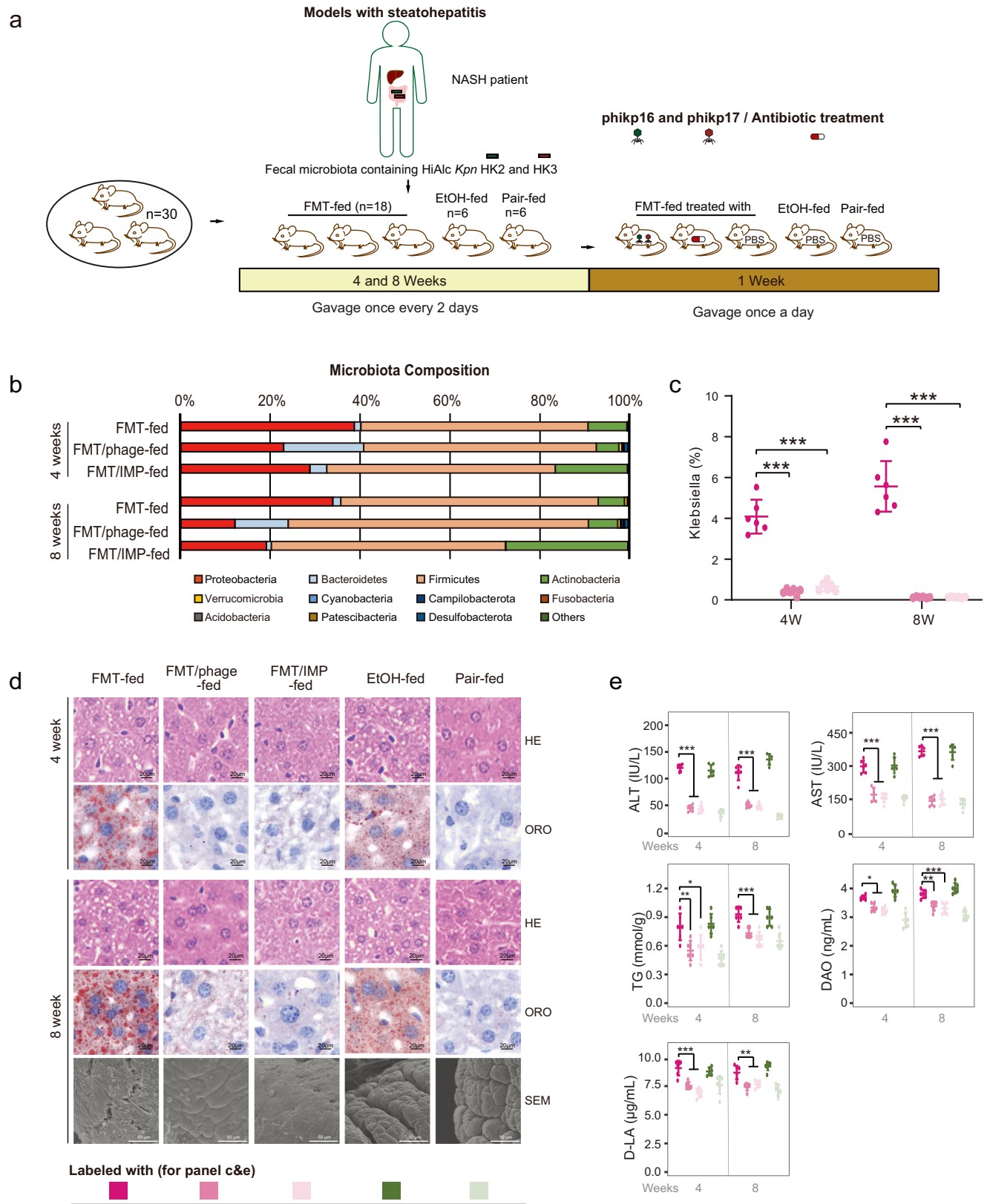

The liver is also an important metabolic organ of lipids and carbohydrates, while excessive accumulation of lipids and gluconeogenesis can promote NAFLD[35,36]. For example, fructose is both an inducer of and a substrate for de novo lipogenesis in the liver, which is responsible for alteration in the communication between the liver and the intestine. In addition to lipogenic effects, inflammation and oxidative stress induced by fructose are important factors for triggering NAFLD[37]. In terms of metabolomics, our data revealed that phage alleviated steatohepatitis by improving metabolism of glycerolipids, glycerophospholipids and various carbohydrates. Phage also downregulated mitochondrial dysfunction related anaerobic glycolysis, which has been thought to be involved in pathogenesis of

**Fig. 7 | Bacteriophage therapy relieved HiAlc *Kpn*-induced steatohepatitis in germfree mice. a** Constructions of FMT-, FMT/phage-, FMT/IMP-, EtOH- and Pair-fed murine models. Mice were treated with FMT, EtOH or YPD broth for 4 weeks or 8 weeks, and then gavaged with phage, antibiotic imipenem or PBS for a further week. After the construction of murine models, mice were euthanized for subsequent analysis. **b** Compositions of phylum-based intestinal microbiota in feces of experimental mice. **c** Abundance of *Klebsiella* in feces of experimental mice (*n* = 6, *P* < 0.0001). **d** HE and ORO stainings of sections of liver tissues of experimental mice, and scanning electron microscope (SEM) micrographs of the proximal colon showing the colonization status of HiAlc *Kpn* in vivo in germfree mice gavaged for 8 weeks. **e** Concentrations of ALT, AST, TG, DAO, and D-LA in serum of experimental mice (*n* = 6, except the *P* value of TG levels between the group FMT-fed and FMT/phage-fed in 4-week model is 0.006, the *P* value of TG levels between the group FMT-fed and FMT/IMP-fed in 4-week model is 0.021, the *P* value of DAO levels between the group FMT-fed and FMT/phage-fed in 8-week model is 0.001, the *P* value of D-LA levels between the group FMT-fed and FMT/phage-fed in 8-week model is 0.001, and the *P* value of D-LA levels between the group FMT-fed and FMT/IMP-fed in 8-week model is 0.004, other *P* values < 0.0001). Values are expressed as the mean ± SD (*n* = 6 mice/group). One-way ANOVA, *P* value < 0.05 (*), 0.01 (**), or 0.001 (***). Source data are provided as a Source data file.

NAFLD[8,9]. A recent study has also shown that phage can impact metabolomic profiles by cascade effects on interbacterial interactions[38]. Our data also showed that the combination of phages had positive effects on inhibiting HiAlc *Kpn* in feces from a patient with NASH. Although phage resistance poses an inevitable threat to the phage application, many studies have demonstrated that using cocktails of phage can mitigate this problem[18,39]. Again, further investigations on the evolution of phage resistance will be needed for treating HiAlc *Kpn*-caused endo-AFLD.

Also worth noting is that the results of NASH model induced by MCD diet. There was only a statistically significant difference in the NAS score between the MCD group and the MCD/HK1 group. Although there was no statistically significant difference in serological indicators, however, the mean values of AST/TG/D-LA/DAO levels of mice in the MCD/ HK1 group were still higher than those of mice in the MCD group. Furthermore, phage therapy did alleviate a certain degree of pathological damage and significantly reduce concentrations of serological indicators in the MCD/HK1 group, suggesting that HiAlc *Kpn* plays a certain role in the MCD/HK1 group constructed in the present study. Mice in both MCD and MCD/HK1 groups had severe steatohepatitis, in which all serological indicators of the mice were significantly higher than those of mice in the HK1-fed or pair-fed group with Chow diet. All these changes indicate that there were formation of liver injury and fatty liver in these experimental mice. Administration of the HiAlc *Kpn* in mice causes fatty liver towards inflammatory infiltration and hepatic fibrosis, much possibly due to the production of ethanol and other metabolites of the bacteria. Even though, obvious changes in serological indicators may be possibly not observed until the disease progressed to a more severe stage. This, of course, needs to be verified in future research.

Taken together, our data showed that treatment with phage specifically eradicating HiAlc *Kpn* alleviated hepatic steatosis without obvious side effects. Beyond reducing the alcohol attack, phage therapy also influenced inflammation, lipid metabolism, and carbohydrates metabolism. Although these data suggest that phage therapy may be an effective and safe alternative to antibiotics, further study remains to be carried on testing whether usage of phages is effective approach for treatment of patients with endo-AFLD. In addition, studies focused on phage cocktail therapies and phage resistance are also required.

## Methods

### Ethics approval and consent to participate
All human participants signed the informed consent form in the present study which was approved by the Research Board of the Ethics Committee of the Capital Institute of Pediatrics and Beijing Chaoyang Hospital (license number 2021320). All participants signed an informed consent form prior to entering the study. All murine experiments were approved by the Medical Ethics Committee of the Capital Institute of Pediatrics and carried out by the licensed individual with license number DWLL2021009.

### Specimens
HiAlc *Kpn* was diagnosed as the causative agent of a male patient (50-60 years old) with NASH accompanied by recurrent pancreatitis in the clinic. In 2019-2020, the patient was diagnosed as the inflammatory phase of NASH for four times, so that we collected 4 fecal samples (the second, fourteenth, sixteenth and seventeenth samples) from inflammatory phases and 14 fecal samples from recovery phases in the absence of antibiotic treatment. The levels of the patient's ALT, AST and GGT were 63, 96 and 138 U/L, respectively (only measured once in time responding to the second sampling). Histological stainings of sections of percutaneous liver biopsy were shown in Fig. S1b.

Further 20 healthy subjects and 16 NASH patients with no alcoholism were recruited, while fecal samples were firstly collected from these adult participants. Each NASH patient received recommendation for weight loss for three months, and the feces of these patients were subjected to collect.

### Fecal microbiota sequencing
Dynamic changes in abundance of intestinal *Klebsiella* and *Klebsiella* phages of the patient were monitored using fecal microbiota sequencing on the Illumina Hi-seq platform (OE Biotech Corp., Shanghai, China), as previously reported[40].

### Isolation and characterization of phage
After culturing *Kpn* strains in YPD broth (contains 2% glucose) for 12 h, the culture supernatants were then collected for determining alcohol production using headspace gas chromatography-mass spectrometry (GC-MS) on Agilent 6850 flame ionization detector platform. The strains produced ≥30 mmol/L alcohol were defined as HiAlc *Kpn*, while the strains produced ≥20 mmol/L alcohol were defined as MedAlc *Kpn*. Bacteriophage was isolated from feces using HiAlc *Kpn* strain HK1 as the host strain. One gram of untreated feces collected from the patient with NASH was suspended in 50 mL of phosphate-buffered saline buffer (PBS) and centrifuged to remove large particles. Then, the supernatant was passed through a 0.22-μm syringe filter, while the filtrate was further co-cultured with the host strain in a double agar overlay plaque assay. The responding phage has been preserved in the China General Microbiological Culture Collection Center (CGMCC), and named as phiW14 (CGMCC No. 23085).

Phage particles were negatively stained with 2% phosphotungstic acid, and used for morphological observation by TEM. Phage phiW14 was added to log-phase bacteria at 0.0001, 0.001, 0.01, 0.1, 1 and 10 in assay of the optimal MOI. For one-step growth experiments, host strain HK1 was infected with phiW14 at the optimal MOI. The burst size of phiW14 was calculated by the ratio of final number of phage PFU to the initial number of bacterial colony-forming unit (CFU). The lytic ability of phiW14 towards host strain HK1 was tested at MOI 0.0001–10. For stability analysis, phage phiW14 was cultured in Luria-Bertani broth with pH 1–14 and temperature 4–80°C, respectively. Genomic DNA of phage phiW14 was extracted using the phenol-chloroform method, as previously described[41]. Whole-genome sequencing of the purified phage DNA was performed on the Illumina NovaSeq platform (Berry Genomics Corp., Beijing, China)[42,43]. Genome annotation was performed using PHASER (http://phaster.ca/) and Prokka software, and a phylogenetic tree based on amino acid sequences of the terminase large subunit was constructed using MEGA software[44,45]. The phage proteome was analyzed on a SCIEX TripleTOF 5600 platform (Beijing

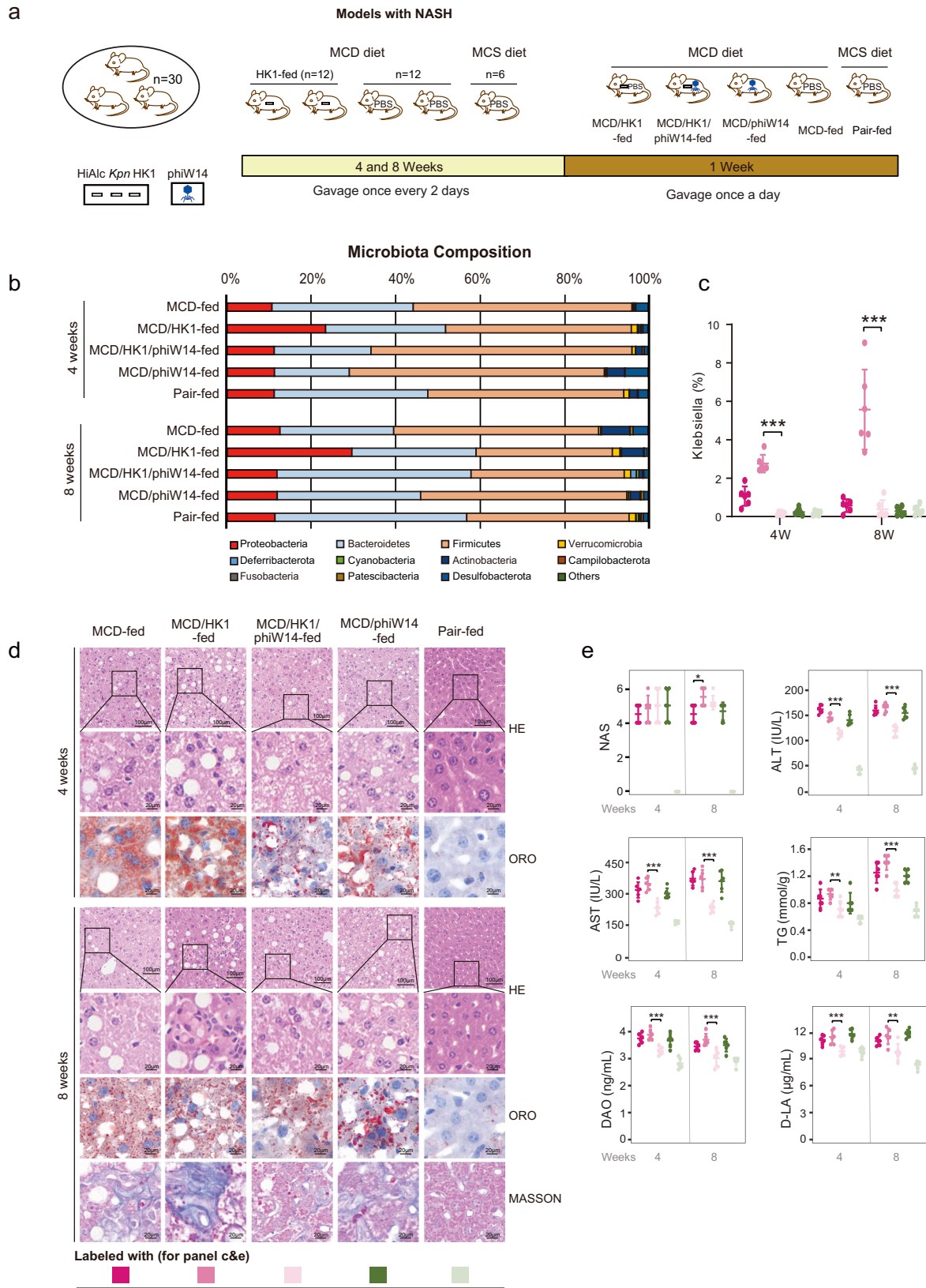

Genomics Institution, Shenzhen, China), while the identified proteins were classified by COG analysis[46].

## Murine models and intervention with medications

Specific-pathogen-free (SPF) male C57BL/6J mice (6-7 weeks old, Charles River Corp., Beijing, China) were used for establishing murine models as previously described[7]. In addition, murine models were also set up using germfree mice. Briefly, germfree C57BL/6J mice were bred under axenic conditions in an isolator (Class Biologically Clean, Madison, WI, USA), and were maintained on a 12:12 h day: night cycle with constant access to food and water. Except murine models of NASH (fed with MCD diet (Research Diets Corp., New Jersey, USA),

**Fig. 8 | The efficacy of bacteriophage therapy in murine model of NASH.**
**a** Constructions of MCD/HK1-, MCD/HK1/phiW14-, MCD/phiW14, MCD- and Pair-fed murine models. Mice were fed with MCD or MCS diet, gavaged with HK1 or PBS for 4 weeks or 8 weeks, and then gavaged with phage or PBS for a further week. After the construction of murine models, mice were euthanized for subsequent analysis. **b** Compositions of phylum-based intestinal microbiota in feces of experimental mice. **c** Abundance of *Klebsiella* in feces of experimental mice ($n = 6$, $P < 0.0001$). **d** HE, ORO and Masson stainings of sections of liver tissues of experimental mice. **e** NAS (Non-alcoholic fatty liver disease activity scores) in liver tissues, and

concentrations of ALT, AST, TG, DAO, and D-LA in serum of experimental mice ($n = 6$, except the $P$ value of TG levels between the group MCD/HK1-fed and MCD/HK1/phiW14-fed in 4-week model is 0.004, and the $P$ value of D-LA levels between the group MCD/HK1-fed and MCD/HK1/phiW14-fed in 8-week model is 0.001, other $P$ values < 0.0001). The murine model was conducted with 3 independent experiments. Values are expressed as the mean ± SD ($n = 6$ mice/group). One-way ANOVA, $P$ value < 0.05 (*), 0.01 (**), or 0.001 (***). Source data are provided as a Source data file.

group of mice fed with MCS (Research Diets Corp., New Jersey, USA) diet was used as negative control. All of the mice were fed with same diet throughout the whole experiment. Natural ingredient NIH #31M Rodent Diet as a standard Chow diet containing 35.17% ground whole wheat, 20% ground whole yellow corn, 10% ground whole oats, 10% gheat middlings, 9% fish meal (60% protein), 5% soybean meal (47.5% protein), 2.5% soy bean oil (no additives), 2% alfalfa meal (17% protein), 2% corn gluten meal (60% protein), 1.5% dicalcium phosphate, 1% brewer's dried yeast, 0.50% ground limestone, 0.50% salt, 0.25% NIH #31 vitamin premix, 0.25% NIH #31 mineral premix, 0.13% choline chloride, 0.10% L-lysine, 0.10% DL-methionine, but without deoxycholate. More details of NIH #31M Rodent Diet can be found in the website (Taconic Biosciences, https://www.taconic.com/quality/animal-diet/). To keep consistent phenotype of experimental mice, all C57BL/6 J mice were also fed the designated forage mentioned as above for 2 weeks prior to experiments. For the HiAlc *Kpn*-induced steatohepatitis, all *K. pneumoniae* strains used in the present study were cultivated in yeast-extract-peptone-dextrose (YPD) broth containing 2% glucose. Mice were inoculated intragastrically once every 2 days with strain HK1 ($10^7$ CFU/200 μL) for 4 weeks or 8 weeks, while mice gavaged with ethanol (40%/ 200 μL, diluted by saline) and YPD broth (200 μL) once every 2 days were used as positive and negative controls, respectively. For FMT mice, fresh feces were collected from a patient with NASH caused by HiAlc *Kpn*. Preparation of the collected feces and garage of mice were performed as described previously[7,47]. Briefly, fresh feces (180 mg) were suspended in 1 mL of PBS for centrifugation at 2000 × $g$ to obtain the bacteria-enriched supernatants, which were continually centrifuged at 15,000 × $g$ for collecting the fecal microbiota. Bacteria pellets were collected and resuspended in YPD broth. Untreated mice were used as recipients and gavaged with $10^9$ CFU of fecal microbiota once per day. In the group FMT/phage, SPF mice were gavaged with fecal microbiota pretreated in vitro with phikp16 and phikp17, while germfree mice were gavaged with fecal microbiota without pretreated with phages. In groups of HK1/phiW14 and HK1/phikp15, mice were gavaged with HiAlc *Kpn* strain HK1 pretreated in vitro with phiW14 and phikp15, respectively. Phage phiW14 ($10^4$, $10^5$ or $10^6$ PFUs) or imipenem (200 μg/200 μL) were used to treat mice with HiAlc *Kpn*-induced steatohepatitis by intragastric administration for 1, 4 or 7 days. All phages used in vivo were purified using Detoxi-Gel™ Endotoxin Removing Gel (Pierce Corp., Rockford, USA) following the manufacturer's instructions.

### Detections of histology, bacterial diversity, serological index, cytokines, and lipid metabolism-associated genes in murine models

Mice were fasted overnight and euthanized. The liver, intestine and kidney tissues of experimental mice were collected and fixed in 10% formalin (pH 7.2) for hematoxylin-eosin and Oil Red O staining (Solarbio Corp., Beijing, China).

To analyze bacterial diversity, total genomic DNA was extracted from murine feces for *16S rRNA* sequencing (OE Biotech). The V3-V4 variable regions of 16S rRNAs were amplified for library construction using primers 343F (TACGGRAGGCAGCAG) and 798R (AGGGTATC-TAATCCT). Raw data were preprocessed using Trimmomatic and QIIME software to remove reads with ambiguous bases, reads with

homologous sequences or sequences <200 bp[48,49]. Acquired clean data were clustered to generate operational taxonomic units (OTUs) using a 97% cutoff value. Representative reads of each OTU were then annotated and blasted in the Silva database[50].

Serum levels of ALT, AST, TC, TG, TBil, DBil, IBil and TBA were measured using a Hitachi 7060 automatic biochemical analyzer. Concentrations of DAO and D-LA were analyzed by enzyme-linked immunosorbent assay (Nanjing Jiancheng Bioengineering Institute, Nanjing, China). Blood samples collected from the portal veins of mice were used for determination of BAC using headspace GC-MS.

Extracted proteins of liver tissues were used to detect cytokines (IL-1α, IL-1β, IL-2, IL-3, IL-4, IL-5, IL-6, IL-9, IL-10, IL-12 p40, IL-12 p70, IL-13, IL-17A, Eotaxin, G-CSF, IFN-γ, CXCL-1, MCP-1, MIP-1α, MIP-1β, RANTES and TNF-α) using a Bio-Plex Pro Mouse Cytokine Grp I Panel on the Bio-Plex MAGPIX System.

Relative transcriptional levels of lipid metabolism-associated genes (lipogenic genes *Lxrα, Lxrβ, Srebp-1c, Fas* and *Scd-1*; lipolytic genes *Pparα, Aco, Lcad, Cyp4a10* and *Cyp4a14*) in liver tissues were determined by real-time quantitative PCR. Total RNA was extracted from liver tissue using Trizol reagent and then reverse transcribed using a PrimeScript™ reagent kit with gDNA Eraser (Takara Corp., Shiga, Japan). The acquired cDNA was detected with SYBR Green I using ABI QuantStudio6 system, while expressions of relative genes were calculated by the $2_T^{-\Delta\Delta C}$ method[51]. The primers (gene *18S* was used as reference gene) used in the present study are listed in Supplementary Data 7.

### Transcriptome sequencing and analysis

The total RNA of liver tissue was extracted using a mirVana miRNA Isolation kit (Ambion Corp., Texas) following the manufacturer's instructions. Samples with RNA integrity number ≥7 were used for library construction. Transcriptome sequencing was performed on the Illumina HiSeq platform (OE biotech). Trimmomatic software was used to remove low-quality and poly-N reads, while hisat2 was used for mapping clean reads to the reference genome[48,52]. DEGs were identified using the DESeq R package; the threshold for significant differential expression was $P$ value < 0.5 and fold change >2 or <0.5. Kyoto Encyclopedia of Genes and Genomes (KEGG) pathway enrichment of DEGs were analyzed using the R package based on the hypergeometric distribution[53].

### Metabolomics analysis

Hepatic metabolomics analysis was performed by GC-MS using an Agilent7890B gas chromatography system and Agilent5977B MSD system (Shanghai Lu-Ming Biotech Corp., Shanghai, China). Significant differential metabolites were determined using partial least-squares-discriminant analysis and orthogonal partial least-squares-discriminant analysis (OPLS-DA). VIP values were used to sort the differential metabolites. Differential metabolites with $P$ value < 0.05 and VIP value > 1 were chosen for Metabo Analyst-based KEGG pathway enrichment analysis[54,55].

### Statistics and reproducibility

Data are presented as the mean ± SD and were subjected to one-way analysis of variance using SPSS 20.0 software (IBM Corp., New York, USA). Difference was judged statistically significant at $P$ value < 0.05

(*), 0.01 (**), or 0.001 (***). All experiments were repeated three times independently with similar result (including the results of Figs. 1a, 2e, 3c, e, 4b, 7d, 8d, and S1b).

## Reporting summary

Further information on research design is available in the Nature Portfolio Reporting Summary linked to this article.

## Data availability

The whole-genome sequence of *K. pneumoniae* phage phiW14 in this study has been deposited in the NCBI under accession code OK655936. The raw sequence data of transcriptome in this study have been deposited in the Genome Sequence Archive under accession code PRJCA008648/CRA006357. The raw sequence data of metabolome in this study have been deposited in the Open Archive for Miscellaneous Data under accession code PRJCA008973/OMIX001072. The bacteria host range, putative open reading frames, identified proteins of phage phiW14, characteristics of clinical NASH patients and healthy controls, differential expression genes of liver tissues in endo-AFLD mice, differential metabolites of feces in endo-AFLD mice, and primers used in this study are provided in the Supplementary Data. Source data are provided with this paper.

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

## Acknowledgements

We thank for the financial support by grants from the National Natural Science Foundation (82002191, C.Y; 82130065, J.Y.; 32170201, G.X), Beijing Natural Science Foundation (7222014, J.Y.), FENG foundation (FFBR 202103, J.Y.), the Research Foundation of Capital Institute of Pediatrics (CXYJ-2021-04, J.Y.), and Public service development and reform pilot project of the Beijing Medical Research Institute (BMR2019-11, J.Y.).

## Author contributions

Conceptualization, J.Y. and L.G.; methodology, L.G., Y.F., B.D., H.F., Z.T., C.Y., G.X., X.C., R.Z., J.C., H.Z., J.F., Z.X., Z.F., T.F., S.D., S.L., Q.Z. and Z.Y.; writing—original draft, L.G.; writing—review and editing, J.Y. and Y.S.; funding acquisition, J.Y. All authors reviewed the manuscript.

## Competing interests

The authors declare no competing interests.
