## [Peer Review File · Nature Communications]

Bacteriophage targeting microbiota alleviates non-alcoholic fatty liver disease induced by high alcohol-producing *Klebsiella pneumoniae*REVIEWER COMMENTS

Reviewer #1 (Remarks to the Author):

This is an interesting study using phage therapy against ethanol-producing *Klebsiella pneumoniae* in a model of diet-induced steatohepatitis. I have a several comments:

1. Current results with phage therapy are recapitulating experiments that the authors have published in their Cell Metabolism paper in 2019. In that paper the authors showed that pretreatment of feces with phages targeting ethanol-producing *Klebsiella pneumoniae* decrease liver disease. To advance these experiments, the authors need to treat mice colonized with stool from NASH patients with phages by oral gavage, which better mimics a human situation. Ideally, the authors should colonize germfree mice and treat with phages by oral gavage.
2. Can the authors please confirm that animal models and results are not the same as in their Cell Metabolism paper?
3. Does the host strain HK1 become resistant during a prolonged incubation (eg 24-48hrs) with phiW14 in vitro? This is an obvious concern if one phage alone is instead of a phage cocktail.
4. Fig. 3 and 4: describe what has been done in more detail for each subpanel in figure legend. How long did the mice receive HK1? 4 or 8 weeks? When was the phage intervention done in relation to HK1 gavage? When were mice harvested in relation to the last phage gavage. The information could additionally be incorporated into Fig. 4A. This should be also better explained in Methods to enhance reproducibility.
5. Number of mice should be stated in the figure legend for each experiment. Abbreviations need to be defined in figure legends, e.g. what does MC or HC stand for in Fig 3?
6. The term NASH/NAFLD is reserved for patients. In mice, the term diet-induced steatohepatitis or fatty liver disease is more appropriate.
7. I could not find a statement about approval of animal experiments by an institutional animal care program.
8. The entire manuscript needs editing for English, grammar and typos.

Reviewer #2 (Remarks to the Author):

In this innovative study, the investigators used phage based knockdown of klebsiella to reduce NAFLD in a mouse model. The innovations and key findings include the observations of individuals with alcohol producing klebsiella having fatty liver disease and the lack of major disruption of the microbiome by introduction of the phages and amelioration of inflammatory pathways based on transcriptomic and metabolomic analyses. These are interpreted to support a causative role for klebsiella in NAFLD and the potential for phage based therapy for NAFLD. There are however several concerns and limitations in this work as well.

Core Concept: There are several concerns about the prevalence and relevance of alcohol producing klebsiella as a cause of NAFLD. While an increase in proteobacteria is common, their causal role is less clear. Several proteobacteria are associated with increased inflammation e.g. Bilophila and the specificity of associations with klebsiella remain controversial. Further, without detailed diet information it is unclear how the microbiome changes occur. Is Deoxycholate involved? Have the authors simply reduced carbohydrates in the diet to remove the substrate for alcohol production? This would have been an obvious next step from the previous publication.

Also, The authors claim that during the inflammatory phase of the index case with NASH, abundance of Klebsiella was increased. It is unclear what this means i.e. during periods of increased ALT, worsened histology..how was this measured..this is not specified in the protocol either.

Methods:

1. The C57Bl6J mouse has an inconsistent NAFLD phenotype and is not an optimal model to study steatohepatitis especially with fibrosis.
2. How also the alcohol production measured in the medium and high alcohol species? What was the correlate in blood alcohol? Was this adjusted for carbohydrate intake?
3. Alcohol feeding generally results in weight loss in mice. Critical details of the alcohol feeding protocol are missing in the MS.
4. The histology is not typical for NASH. Even mouse NAFLD. It would be ideal to show 10x view so that the overall architecture can be evaluated. There is no trichrome stain. The Oil Red O is not sufficient to establish NASH.
5. Under typical NASH diets used in mice, it would be imperative to show that knockdown of klebsiella reduces the disease phenotype. Neither condition has been met to establish causality.

Based on these, the interpretation of this work and its translatability to the human state are debatable.

Reviewer #1 (Remarks to the Author):

This is an interesting study using phage therapy against ethanol-producing *Klebsiella pneumoniae* in a model of diet-induced steatohepatitis. I have a several comments:

1. Current results with phage therapy are recapitulating experiments that the authors have published in their Cell Metabolism paper in 2019. In that paper the authors showed that pretreatment of feces with phages targeting ethanol-producing *Klebsiella pneumoniae* decrease liver disease. To advance these experiments, the authors need to treat mice colonized with stool from NASH patients with phages by oral gavage, which better mimics a human situation. Ideally, the authors should colonize germfree mice and treat with phages by oral gavage.

Response:

We thank the reviewer for the constructive suggestion. Following the suggestion, we established models of NAFLD in germfree mice gavaged with HiAlc *Kpn* strains HK2 (ST447, producing 42.5 mmol/L alcohol) and HK3 (ST101, producing 41.5 mmol/L alcohol) isolated from a patient with NASH. The phages phikp16 and phikp17 were isolated from the same patient's feces, which specifically targeted HK2 and HK3, *in vitro* respectively. For FMT, fecal samples of the patient mentioned above were conducted to be transplanted to the experimental mice. The results indicated that gavage of intestinal flora from the patient with NASH could lead steatohepatitis in models of germfree mice. In the meanwhile, *Kpn* colonizations in gut of the experimental mice were observed using SEM. After treatment with phages phikp16 and phikp17, symptoms of steatohepatitis were alleviated, while the *Kpn* strains were eliminated from the gut of experimental mice, which was confirmed by using SEM and 16S *rRNA* sequencing. We have added the supplemented data derived from models of germfree mice with NAFLD in line 231-239 and Figure 7 of the revised manuscript.

Figure Bacteriophage therapy relieved HiA1c *Kpn*-induced steatohepatitis in germfree mice.

(A) Constructions of FMT-, FMT/phage-, FMT/IMP-, EtOH- and Pair-fed murine models. Mice were treated with FMT, EtOH or YPD broth for 4 weeks or 8 weeks, and then gavaged with phage, antibiotic imipenem or PBS for a further week. After the construction of murine models, mice were euthanized for subsequent analysis. (B) Compositions of phylum-based intestinal microbiota in feces of experimental mice. (C) Abundance of *Klebsiella* in feces of experimental mice. (D) HE and ORO stainings of sections of liver tissues of experimental mice, and scanning electron microscope (SEM) micrographs of the proximal colon showing the colonization status of HiA1c *Kpn* *in vivo* in germfree mice gavaged for 8 weeks. (E) Concentrations of ALT, AST, TG, DAO, and D-LA in serum of experimental mice. Values are expressed as the mean \pm SD ($n \geq 6$ mice/group). One-way ANOVA, P-value < 0.05 (*), 0.01 (**), or 0.001 (***).

2. Can the authors please confirm that animal models and results are not the same as in their Cell Metabolism paper?

Response:

We thank for the reviewer's concerns. The differences of animal models between the present manuscript and our previous paper published in Cell Metabolism are as follows:

(1) Difference in objective. The present study is focusing on the effect of phage therapy on mice of NAFLD, while our previous study was intended to demonstrate that HiAlc *Kpn* is the bacterial culprit of endo-AFLD;

(2) Difference in methods. In our previous study, HiAlc *Kpn* or microbiota for endo-AFLD model construction were pretreated with phage *in vitro*. In the present study, mice were first gavaged with untreated HiAlc *Kpn* or microbiota and then treated with phage *in vivo*. Furthermore, we also optimized the concentration and therapeutical time of phage therapy used in the animal models;

(3) Difference in *Kpn* strains used for construction of animal models. In the study published by Cell Metabolism, strains used in our studies were HiAlc *Kpn*, strains W14 and TH1 (producing 63.2 and 60.8 mmol/L alcohol, respectively) were isolated from an individual with severe NASH accompanied by auto-brewery syndrome (ABS). In the present study, strain HK1 was initially isolated from a patient with NASH accompanied by recurrent pancreatitis, which produces 49.7 mmol/L alcohol, although all these strains HK1, W14 and TH1 belong to HiAlc *Kpn*;

Furthermore, the present study suggests that phage could alleviate symptoms of endo-AFLD, including the relieved hepatic dysfunction and the improved expressions of cytokines and lipogenic genes *in vivo*. In addition to reduce alcohol attack, the results of the present study showed that phage therapy also influenced inflammation, and lipid and carbohydrate metabolism. In our previous study, however, all results regarding usage of phage only verified that phage has the ability to specifically inhibit the relevant bacteria *in vitro*.

3. Does the host strain HK1 become resistant during a prolonged incubation (eg 24-48hrs) with phiW14 *in vitro*? This is an obvious concern if one phage alone is instead of a phage cocktail.

Response:

We thank for the reviewer's concern. Indeed, although many host strains could become resistant

during a prolonged incubation (eg 24-48hrs) with phages, there was no resistant mutant during a prolonged incubation in the present study. Even though, we entirely agree with the reviewer's suggestion for using phage cocktail. In the further study of clinical application, we will try to use phage cocktail to combat resistance.

4. Fig. 3 and 4: describe what has been done in more detail for each subpanel in figure legend. How long did the mice receive HK1? 4 or 8 weeks? When was the phage intervention done in relation to HK1 gavage? When were mice harvested in relation to the last phage gavage. The information could additionally be incorporated into Fig. 4A. This should be also better explained in Methods to enhance reproducibility.

Response:

Many thanks for the reviewer's suggestion. We have added these information in figure legends in the revised manuscript as follows:

(1) Fig.3. Mice with steatohepatitis (gavaged with HiAlc *Kpn* for 4 weeks) were treated with phage phiW14 at the range of 10^4 (low concentration, LC), 10^5 (median concentration, MC), and 10^6 (high concentration, HC) PFUs for 1, 4 and 7 days (line 663-665);

(2) Fig.4. Mice were gavaged with strain HK1, EtOH or YPD broth for 4 weeks or 8 weeks, and then were treated with phage phiW14, antibiotic imipenem or PBS for a further week. After the construction of models, mice were euthanized for subsequent analysis (line 695-698).

5. Number of mice should be stated in the figure legend for each experiment. Abbreviations need to be defined in figure legends, e.g. what does MC or HC stand for in Fig 3?

Response:

Again, many thanks for the reviewer's suggestions. Corresponding to above comments, mice with steatohepatitis (gavaged with HiAlc *Kpn* for 4 weeks) were treated with phage phiW14 at the range of 10^4 (low concentration, LC), 10^5 (median concentration, MC), and 10^6 (high concentration, HC) PFUs for 1, 4 and 7 days. Values are expressed as the mean \pm SD of ≥ 3 independent experiments ($n \geq 6$ mice/group/experiment). * $P < 0.05$, ** $P < 0.01$ or *** $P < 0.001$. We have added these definitions in line 663-673 of the revised manuscript.

6. The term NASH/NAFLD is reserved for patients. In mice, the term diet-induced steatohepatitis or fatty liver disease is more appropriate.

Response:

We thank the reviewer for the helpful suggestion. We have revised “NASH/ NAFLD” of mice to “steatohepatitis” throughout the manuscript.

7. I could not find a statement about approval of animal experiments by an institutional animal care program.

Response:

Thanks for the reviewer’s comments. All murine experiments were approved by the Medical Ethics Committee of the Capital Institute of Pediatrics and carried out by the licensed individual with license number DWLL2021009. We have added this part in line 333-335 in the revised manuscript.

8. The entire manuscript needs editing for English, grammar and typos.

Response:

Thanks for the reviewer’s suggestion. The new version of manuscript has been edited by a qualified native English-speaking professor.

Reviewer #2 (Remarks to the Author):

In this innovative study, the investigators used phage based knockdown of klebsiella to reduce NAFLD in a mouse model. The innovations and key findings include the observations of individuals with alcohol producing klebsiella having fatty liver disease and the lack of major disruption of the microbiome by introduction of the phages and amelioration of inflammatory pathways based on transcriptomic and metabolomic analyses. These are interpreted to support a causative role for klebsiella in NAFLD and the potential for phage based therapy for NAFLD. There are however several concerns and limitations in this work as well.

1. Core Concept: There are several concerns about the prevalence and relevance of alcohol producing klebsiella as a cause of NAFLD. While an increase in proteobacteria is common, their causal role is less clear. Several proteobacteria are associated with increased

inflammation e.g. Bilophila and the specificity of associations with klebsiella remain controversial. Further, without detailed diet information it is unclear how the microbiome changes occur. Is Deoxycholate involved? Have the authors simply reduced carbohydrates in the diet to remove the substrate for alcohol production? This would have been an obvious next step from the previous publication.

Response:

We thank the reviewer for the critical comments. We entirely agree with the reviewer's opinion, that is, the disease could be caused by a combination of many bacteria, and closely related to diet. In fact, the causal relationship between intestinal flora and disease is indeed complex, which does need to be further studied. It has been shown that several *Proteobacteria* bacteria might associate with intestinal inflammation. Although numerous studies have been performed, which bacteria is the culprit is always difficult to be determined. In this point of view, we initially isolated HiAlc *Kpn* from an extreme case suffered from ABS and NASH, proposed and tested the hypothesis of endo-AFLD (Jing Yuan, *Cell Metab* 2019, PMID 31543403).

(1) For the prevalence and relevance of alcohol-producing *Klebsiella* as a cause of NAFLD, in the present study, in addition to proving the conclusion on animal models, we also verified this in a cohort.

In the study published by Cell Metabolism, we have previously found that the distributions of the phylum *Proteobacteria* and species *Kpn* strongly correlated with the fluctuations in blood alcohol concentration in consecutive feces collected from an individual with NASH and ABS during the pre-onset, onset, recovery, and post-treatment stages (Fig. 1A of the previous paper, which were shown as Fig. a, please see below). Furthermore, the results of our cohort showed that 61% of NAFLD patients carried HiAlc and MedAlc *Kpn*, while this value was only 6.25% in controls (Fig. 1F and G of the previous paper, which were shown as Fig. b and Fig. c below). Taken together, our results showed that HiAlc *Kpn* might be strong associated with NAFLD (Jing Yuan, *Cell Metab* 2019, PMID 31543403). Similarly, the present manuscript also showed that the quantity of *Kpn* in NASH patients was significantly higher than these of the same followed-up patients (after weight loss) and healthy controls, while the content of *Klebsiella* phages was just the opposite to *Kpn* (Fig. S1B of the present manuscript, which was shown as Fig. d below);

Figure Commensal HiAlc *Kpn* has a higher statistical chance of initiating NAFLD.

(a) Correlations of the intestinal microbiota and blood alcohol concentration (BAC) of NASH and ABS patients. The gut microbiota compositions of the samples are presented according to the BAC (from high to low). The percentages of *Proteobacteria* and *Klebsiella* are outlined; (b) Relative concentrations of *Kpn* and *Klebsiella* phages in fecal samples collected from NASH patients and NASH patients after weight loss and controls. Values are expressed as the mean \pm SD. One-way ANOVA, P -value < 0.05 (*), 0.01 (**) or 0.001 (***); (c) The alcohol concentrations measured in fermented fecal samples from NAFLD patients (carmine box), NAFL (orange box), NASH (blue box), and controls (brown box); (d) Alcohol-producing ability measured for the highest alcohol-producing *Kpn* isolates.

(2) All mice in each group were fed with the same diet throughout the whole experiment. Natural ingredient NIH #31M Rodent Diet was used as a standard Chow diet containing 35.17% ground whole wheat, 20% ground whole yellow corn, 10% ground whole oats, 10% gheat middlings, 9% fish meal (60% protein), 5% soybean meal (47.5% protein), 2.5% soy bean oil (no additives), 2% alfalfa meal (17% protein), 2% corn gluten meal (60% protein), 1.5% dicalcium phosphate, 1% brewer's dried yeast, 0.50% ground limestone, 0.50% salt, 0.25% NIH #31 vitamin premix, 0.25% NIH #31 mineral premix, 0.13% choline chloride, 0.10% L-lysine, 0.10% DL-methionine, but without deoxycholate. More details of NIH #31M Rodent Diet can be found in the website (Taconic

Biosciences, <https://www.taconic.com/quality/animal-diet/>). We have added the information of diet in line 388-396 of the revised manuscript.

2. Also, The authors claim that during the inflammatory phase of the index case with NASH, abundance of Klebsiella was increased. It is unclear what this means i.e. during periods of increased ALT, worsened histology.. how was this measured.. this is not specified in the protocol either.

Response:

We thank the reviewer for the critical comments. We collected 4 fecal samples from inflammatory phases of the patient in the absence of antibiotic treatment and measured the abundance of *Kpns*. In the meantime, we also measured blood levels of liver enzymes and analyzed histological features of percutaneous liver biopsy. The concentrations of ALT, AST and GGT was 63, 96 and 138 U/ L, respectively; Histological stainings (HE, Reticular fiber and Masson) showed that there were increases in fatty droplets, infiltration of inflammatory cells and deposition of collagens in sections of the liver tissues, suggesting that NASH patient was in liver fibrosis stage (please see Figure below). We have added the results of ALT, AST and GGT in line 341-344, and added the histological stainings into the Fig. S1B of the revised manuscript.

Figure Histological stainings (HE, Reticular fiber and Masson) of percutaneous liver biopsy in the patient with NASH.

Methods:

1. The C57Bl6J mouse has an inconsistent NAFLD phenotype and is not an optimal model to study steatohepatitis especially with fibrosis.

Response:

Thanks for the reviewer's concerns. The C57BL/6J mouse is indeed not the only proper animal

model for NAFLD study. Actually, we initially constructed both C57BL/6J and Balb/c models of NAFLD simultaneously. The C57BL/6J mouse model showed good adaptability throughout the whole experiment. However, death frequently occurred in Balb/c models because of the less tolerance to alcohol and HiAlc *Kpn* feeding. To keep consistent phenotype of experimental mice, all C57BL/6J mice were also fed with the designated forage mentioned in the present manuscript for 2 weeks before experiment (we have added this information in line 396-398 of the revised manuscript).

In addition, it has been shown that there are numerous studies using C57BL/6J mice as model for NAFLD study (for example, Yi Duan, *Nature* 2019, PMID 31723265; Meng Li, *Gut* 2018, PMID 28877979; Olivier Govaere, *J Hepatol* 2022, PMID 34942286; Xiaoya Li, *Nucleic Acids Res* 2020, PMID 32710621). Thus, we finally chose C57BL/6J mouse models for NAFLD study in our previous study (Jing Yuan, *Cell Metab* 2019, PMID 31543403) and the present study. For the rigor of research, we will try more animal models for further verification in the future.

2. How also the alcohol production measured in the medium and high alcohol species? What was the correlate in blood alcohol? Was this adjusted for carbohydrate intake?

Response:

We thank the review for the concerns.

(1) In our previous research, by used *Escherichia coli*, *Shigella*, *Salmonella* and other *Enterobacteriaceae* as controls, we measured the alcohol concentration after culture under the same conditions. According to our previous published data, HiAlc and MedAlc *Kpn* were defined as ≥ 30 mmol/L and ≥ 20 mmol/L in production of alcohol, respectively (Jing Yuan, *Cell Metab* 2019, PMID 31543403; Nannan Li, *Gut Microbes*, 2021, PMID 34632939).

The details regarding “the alcohol production measured” included two aspects, and the details are as follows:

- i. Regarding “the alcohol production measured” in strains: In the *in vitro* experiments, the alcohol production through the fermentation by these strains was very obvious. In order to demonstrate the high alcohol producing ability of the strains, the strains were cultured in YPD broth (contains 2% glucose) for 12 h. After culturing, the supernatants were collected for determining alcohol production using headspace gas chromatography-mass spectrometry (GC-MS) on

Agilent 6850 flame ionization detector platform (please see line 354-356 in Method);

- ii. Regarding “the alcohol production measured” in mice: C57BL/6J SPF mice were fed with normal chow diet, then randomly divided into five groups, in which three of groups were gavaged with $\sim 10^7$ CFU of HiAlc *Kpn* (suspended in 200 μ L YPD medium); the rest two groups including mice gavaged with a single dose of ethanol (40% ethanol, 200 μ L) and pair-fed mice gavaged with YPD medium (200 μ L) were used as positive and negative controls, respectively. After the models were constructed by gavaging with each reagent once every two days for 4 weeks or 8 weeks, phage phiW14 (10^4 , 10^5 or 10^6 PFUs) and imipenem (200 μ g/ 200 μ L) were used to treat two groups of mice gavaged with HiAlc *Kpn* for a further week, respectively. The gavage was always performed in the early morning. In this case, 100% of mice survived after feeding with strains or ethanol. The mice were always euthanized for 9 h post gavage. Number of animals for each subpanel was ≥ 6 , while number of experiments was ≥ 3 (please see line 381-414 in Method). In addition, BAC (collected from the portal veins) of mice were determined by using headspace GC-MS (please see line 433-434 in Method);

We have added the method of alcohol concentration detection into line 354-356 and 433-434 of the revised manuscript;

(2) In addition, we did not adjust for carbohydrate intake during the whole experiment;

(3) According to the existing results, concentrations of blood alcohol could be higher in HiAlc *Kpn*-feeding mice than that of MedAlc *Kpn*-feeding mice;

Under normal conditions, ethanol is produced constantly by the intestinal microbiota in the human gut. However, endogenous ethanol is removed rapidly and almost completely from portal blood by liver alcohol dehydrogenases (ADHs), catalases, and the microsomal ethanol-oxidizing system. Thus, the ability of gut microbiota to produce hepatotoxic concentrations of endogenous ethanol in the clinical diagnosis was concealed because of a negligible and undetected alcohol concentration in blood.

In our previous experiments, ethanol concentrations are more than 2-folds higher in portal vein blood as compared to that of peripheral blood of the colonized mice. Therefore, the relationship between the concentration of alcohol produced by bacteria in the culture medium and the concentration of alcohol in the blood might be varied from individuals, which is difficult to be quantified. Even though, it is certainly worth to further explore the internal relationship and

mechanism between alcohol concentrations in the culture medium and in blood of mice as far.

3. Alcohol feeding generally results in weight loss in mice. Critical details of the alcohol feeding protocol are missing in the MS.

Response:

Many thanks for the reviewer's suggestions. Actually, we did measure the weight of the experimental mice (please see Figure below), in which mice gavaged with ethanol (40%/ 200 μ L, diluted by saline) once every 2 days were used as positive control (the alcohol feeding protocol has been added in line 401-402 of the revised manuscript).

The results of the weight of mice showed that both alcohol and HiA1c *Kpn* feeding resulted in weight loss in murine models ($P < 0.05$). Interestingly, although there was no statistically significant difference in body weight between HK1- and HK1/phiW14-fed groups, the average weight of group HK1/phiW14-fed (25.83 g in 4 weeks, 28.14 g in 8 weeks) was also higher than that of group HK1-fed (25.40 g in 4 weeks, 27.02 g in 8 weeks). These suggest that phage treatment might improve health of mice to a certain extent. The extra data have added into the Results in line 192-198 and Fig.4F of the revised manuscript.

Figure Body weight of experimental mice. Values are expressed as the mean \pm SD ($n \geq 6$ mice/group). One-way ANOVA, P -value < 0.05 (*), 0.01 (**), or 0.001 (***)).

4. The histology is not typical for NASH. Even mouse NAFLD. It would be ideal to show 10x view so that the overall architecture can be evaluated. There is no trichrome stain. The Oil Red O is not sufficient to establish NASH.

Response:

Thanks for the reviewer's comments.

(1) We have added the 10x view of pathological microphotographs into the revised Fig. 4B;

Figure HE and ORO stainings of sections of liver tissues of experimental mice.

(2) In our previous study (Jing Yuan, *Cell Metab* 2019, PMID 31543403), only feeding both with high-fat diet (containing 60% fat, New Brunswick, NJ) and HiAlc *Kpn*, but not with chow diet (mentioned above) plus HiAlc *Kpn* could lead to obvious fibrosis in the experimental mice at 8 weeks as shown in the following figure published by Cell Metabolism:

Figure Liver histology for the assessment of hepatic steatosis in SPF mice fed with high-fat diet for 4 and 8 weeks (Jing Yuan, *Cell Metab* 2019, PMID 31543403).

In the present study, after fed for 4 and 8 weeks with Chow diet, we did perform the Masson staining (please see the stainings below). Similar to the observations in our previous study, there was no obvious fibrosis in sections of the liver tissues of mice fed with HiAlc *Kpn* or EtOH (please see microphotographs below). This is why we did not display the imaging of Masson staining in the present manuscript.

Figure Masson staining of sections of liver tissues of experimental mice.

Following the reviewer's suggestion of Q5, we have added the results of Masson staining into supplementary NASH models.

5. Under typical NASH diets used in mice, it would be imperative to show that knockdown of klebsiella reduces the disease phenotype. Neither condition has been met to establish causality. Based on these, the interpretation of this work and its translatability to the human state are debatable.

Response:

We entirely agree with the reviewer's comments. We have supplemented the NASH models in line 240-251 and Figure 8 in the revised manuscript (please also see below). The results showed that phage treatment was able to alleviate steatohepatitis in mice fed with MCD diet (A02082002BR, Research Diets Corp., New Jersey, USA) plus HiAlc *Kpn* but not in mice with MCD diet alone.

This is probably pointing that there are diversities in the pathogenic mechanisms of NAFLD. On the other hand, these data also suggest that phage therapy might be highly specific, at least in our animal models of NAFLD. It is precisely because of the highly specificity, the existing phage therapies used in the clinic are only suitable these patients carrying the relevant target bacteria. Furthermore, developing rapid, non-invasive methods of HiAlc *Kpn* detection, such as the blood alcohol concentration and the specific gene screening of HiAlc *Kpn* are undergoing in our lab, which may contribute to the clinical applications, including phage therapy.

Figure 8. The efficacy of bacteriophage therapy in murine model of NASH. (A) Constructions of MCD/HK1-, MCD/HK1/phW14-, MCD/phW14, MCD- and Pair-fed murine models. Mice were fed with MCD or MCS diet, gavaged with HK1 or PBS for 4 weeks or 8 weeks, and then gavaged with phage or PBS for a further week. After the construction of murine models, mice were euthanized for subsequent analysis. (B) Compositions of phylum-based intestinal microbiota in feces of experimental mice. (C) Abundance of *Klebsiella* in feces of experimental mice. (D) HE, ORO and Masson stainings of sections of liver tissues of experimental mice. (E) Concentrations of ALT, AST, TG, DAO, and D-LA in serum of experimental mice. Values are expressed as the mean \pm SD ($n \geq 6$ mice/group). One-way ANOVA, P-value < 0.05 (*), 0.01 (**), or 0.001 (***).

REVIEWER COMMENTS

Reviewer #1 (Remarks to the Author):

I have reviewed the revised version of this manuscript. The authors have nicely described now the effect of phage treatment in mice colonized with stool from NASH patients. I still have some comments:

1. The authors still did not answer my question about rigor and reproducibility. The exact number of mice and technical replicates (independent mouse cohorts or experiments) should be stated in the figure legend for each experiment. It is not sufficient to state >6 mice. In addition, I also realized that the bar indicating statistical significance between 2 groups is not aligned with the columns, as far as I can tell. This should be corrected in the figures.

2. I also need to comment on the MCD diet model and its results. It is very surprising that HK1 is not exacerbating disease in MCD-fed mice. This is very concerning and against the hypothesis of the authors. MCD leads to inhibition of VLDL secretion from hepatocytes and to hepatic steatosis. Any additional insult from the gut through ethanol production should exacerbate disease. The authors might have to try a more physiological model with western diet.

Reviewer #2 (Remarks to the Author):

I believe the authors have satisfactorily addressed my concerns. I have no further critiques. My only suggestion is to tone down any implications re NASH and disease progression since you did not have fibrotic NASH based on the Sirius Red stains and no convincing Ballooning or MD bodies.

Reviewer #1 (Remarks to the Author):

I have reviewed the revised version of this manuscript. The authors have nicely described now the effect of phage treatment in mice colonized with stool from NASH patients. I still have some comments:

1. The authors still did not answer my question about rigor and reproducibility. The exact number of mice and technical replicates (independent mouse cohorts or experiments) should be stated in the figure legend for each experiment. It is not sufficient to state >6 mice. In addition, I also realized that the bar indicating statistical significance between 2 groups is not aligned with the columns, as far as I can tell. This should be corrected in the figures.

Response:

We thank the reviewer for the constructive suggestion. We have revised the mice number to “n=6” in figure legends (Figure 2-4, and 6-8), and realigned the bars between columns in figures 2-4, and 7-8 in the new revised manuscript.

2. I also need to comment on the MCD diet model and its results. It is very surprising that HK1 is not exacerbating disease in MCD-fed mice. This is very concerning and against the hypothesis of the authors. MCD leads to inhibition of VLDL secretion from hepatocytes and to hepatic steatosis. Any additional insult from the gut through ethanol production should exacerbate disease. The authors might have to try a more physiological model with western diet.

Response:

We thank for the reviewer’s concern.

(1) We are very sorry for our negligence in not including the panoramic pathological images of HE stainings and not selecting representative fields of view, which resulted in a lack of rigor in our manuscript. We have added the 10x view of pathological microphotographs into the revised Figure 8 (please also see Figure below). From the pathological images, we hope the reviewer will agree with us, i.e. that the MCD/HK1 group had more severe changes in hepatic steatosis and inflammation compared to the MCD group, and that there was a noticeable occurrence of cellular phagocytosis in the MCD/HK1 group of 8-week model.

Figure HE, ORO and Masson stainings of sections of liver tissues of experimental mice.

Besides, we have added pathological scores on the original pathological images in Figure 8e (please also see Figure below). The NAFLD activity scores (NAS, which was evaluated by semiquantitative analysis of steatosis, lobular inflammation, hepatocellular ballooning, and fibrosis) showed that the MCD/HK1 group had a significantly higher score than that of the MCD group in 8-weeks model ($P < 0.05$).

Figure NAS analysis of liver tissues of experimental mice.

Although there were obvious pathological differences between the MCD group and the MCD/HK1 group, there was no statistically significant difference in serum indicators. This is possibly due to that though the hepatic steatosis in both two groups has pathologically occurred, little change in the

serum indicators can be observed until the disease progressed to a more severe stage. In this case, it is a bit arbitrarily to conclude that there is no difference in liver damage between the two groups. This, of course, needs to be verified in further research.

(2) Regarding serum indicators, mice of the bacteriophage-treated MCD/HK1/phiW14 group had significantly lower levels of these indexes measured than that of mice before bacteriophage-treatment, although these did not recover to normal levels (please see Figure below). We believe that this status also might reflect the important role of HiAlc *Kpn* in the pathogenic process of NASH.

Figure Concentrations of serum indicators and NAS analysis of experimental mice.

We have added the 10x view of pathological microphotographs and NAS analysis in the revised Figure 8, and added the detail description of pathological results in line 243-254.

Reviewer #2 (Remarks to the Author):

I believe the authors have satisfactorily addressed my concerns. I have no further critiques.

My only suggestion is to tone down any implications re NASH and disease progression since

you did not have fibrotic NASH based on the Sirius Red stains and no convincing Ballooning or MD bodies.

Response:

Thanks for the reviewer's comments. In addition to revising "NASH/ NAFLD" to "steatohepatitis" or "hepatic steatosis" in the previous revision, the following modifications were also made in the new revised manuscript:

(1) We have revised the conclusion to "phage was able to alleviate steatohepatitis caused by HiAlc *Kpn*" in line 31 and line 327;

(2) We have deleted the descriptions of NASH or disease progression in RESULTS and DISCUSSION of the new revised manuscript.

REVIEWERS' COMMENTS

Reviewer #1 (Remarks to the Author):

My question about independent experiments was not answered. If an experiment with n=6 mice was performed once, it is not sufficient.

Second, there is no significant difference between MCD and MCD/HK1 in NAS, AST, ALT, hepatic TG in Fig. 8e, indicating that the authors cannot reproduce the effect of HK1 in a second model of liver disease.

Reviewer #1 (Remarks to the Author):

1. My question about independent experiments was not answered. If an experiment with n=6 mice was performed once, it is not sufficient.

Response:

We thank for the reviewer's concern. Actually, the NASH model was performed with 3 independent experiments (n = 6 mice/group/per experiment). We have added the information into the figure legends (line 825-827) as follows: The murine model was conducted with 3 independent experiments. Values are expressed as the mean \pm SD (n=6 mice/group).

2. Second, there is no significant difference between MCD and MCD/HK1 in NAS, AST, ALT, hepatic TG in Fig. 8e, indicating that the authors cannot reproduce the effect of HK1 in a second model of liver disease.

Response:

We thank the reviewer for the critical comments.

(1) Indeed, our data showed that there was a statistically significant difference in the NAS score between the MCD group and the MCD/HK1 group.

(2) Although there was no statistically significant difference in serological indicators, the mean values of AST/TG/D-LA/DAO levels in the MCD/HK1 group were still higher than those of the MCD group. Furthermore, phage therapy did alleviate a certain degree of pathological damage and significantly reduce concentrations of serological indicators in the MCD/HK1 group. These suggest that HiAlc *Kpn* did play a certain role in the MCD/HK1 group constructed in the present study. Based on the analysis of our results, we think that some possible reasons for the minimal differences observed between two groups are as follows:

i) Mice in both MCD and MCD/HK1 groups had severe steatohepatitis, in which all serological indicators of the mice were significantly higher than those of mice in the HK1-fed or pair-fed group with Chow diet. All these changes indicate that there were formation of liver injury and fatty liver in these experimental mice. Administration of the HiAlc *Kpn* in mice causes fatty liver towards inflammatory infiltration and hepatic fibrosis, much possibly due to the production of ethanol and other metabolites of the bacteria. Even though, obvious changes in serological indicators may be

possibly not observed until the disease progressed to a more severe stage. This, of course, needs to be verified in future research.

ii) Under the premise of not affecting the main conclusions of the present study, we have incorporated the aforementioned possible reasons into the conclusion and discussion sections (line 36 and 330-344), and emphasized that phage therapy can only partially alleviate NASH induced by other causes combined with HiAlc *Kpn* infection, and might be possibly only applicable to HiAlc *Kpn* -induced endo-AFLD.